# Natural variation of STKc_GSK3 kinase TaSG-D1 contributes to heat stress tolerance in Indian dwarf wheat

Jie Cao[1,2], Zhen Qin [1,2], Guangxian Cui[1], Zhaoyan Chen[1], Xuejiao Cheng[1], Huiru Peng [1], Yingyin Yao[1], Zhaorong Hu [1], Weilong Guo [1], Zhongfu Ni [1], Qixin Sun [1] & Mingming Xin [1] ✉

Heat stress threatens global wheat (*Triticum aestivum*) production, causing dramatic yield losses worldwide. Identifying heat tolerance genes and comprehending molecular mechanisms are essential. Here, we identify a heat tolerance gene, *TaSG-D1^{E286K}*, in Indian dwarf wheat (*Triticum sphaerococcum*), which encodes an STKc_GSK3 kinase. *TaSG-D1^{E286K}* improves heat tolerance compared to *TaSG-D1* by enhancing phosphorylation and stability of downstream target TaPIF4 under heat stress condition. Additionally, we reveal evolutionary footprints of *TaPIF4* during wheat selective breeding in China, that is, InDels predominantly occur in the *TaPIF4* promoter of Chinese modern wheat cultivars and result in decreased expression level of *TaPIF4* in response to heat stress. These sequence variations with negative effect on heat tolerance are mainly introduced from European germplasm. Our study provides insight into heat stress response mechanisms and proposes a potential strategy to improve wheat heat tolerance in future.

Global warming seriously threatens food security worldwide by reducing crop productivity and quality. Wheat grain yield is predicted to decline by approximately 6% for each 1 °C increase in average temperature above the optimum[1], and heat stress has caused a 5.5% reduction in global wheat production over the past three decades[2]. Indian dwarf wheat (*Triticum sphaerococcum*) was endemic to India and Pakistan, and was cultivated for thousands of years before the Green Revolution[3,4]. *T. sphaerococcum* is favored by local farmers due to its enhanced adaptability to local environmental constraints, including high temperature[5], however, the underlying mechanism of heat tolerance is largely unknown. Unraveling the key genes controlling heat tolerance in *T. sphaerococcum* and deciphering their related regulatory network are of great importance to explore this germplasm for the heat tolerance improvement in crops.

In this work, we identify a heat tolerance gene *TaSG-D1^{E286K}* in Indian dwarf wheat and reveal how the gene enhances heat tolerance. In addition, we find that the distribution of *TaSG-D1^{E286K}* is limited to

India and Pakistan, whereas its downstream target *TaPIF4* is evolutionarily selected during Chinese wheat breeding programs.

## Results

### Identification of heat tolerance gene *TaSG-D1^{E286K}* in wheat

We demonstrated that the *T. sphaerococcum* introgression line ND4332 exhibits greater heat tolerance than the *T. aestivum* line HS2, with 78.3% vs.5.7% of seedling survival rate under heat stress conditions (SSRH), respectively (Fig. 1a, b). To characterize the causal genetic loci contributing to heat tolerance, we performed QTL mapping using an $F_7$ recombinant inbred line (RIL) population generated from ND4332 and HS2[6]. Two candidate QTLs associated with heat tolerance were identified using winQTLcart (v2.5) software with a default LOD value cutoff of 3.5, which were located on Chromosome 3D and 5 A, respectively. Of the two QTLs, the genetic interval residing on chromosome 3D attracted our attention, as the ND4332-derived allele confers improved heat tolerance and explained 11.7% of phenotypic variation. This genetic interval is located between markers

[1]Frontiers science center for molecular design breeding, Key Laboratory of Crop Heterosis Utilization (MOE), China Agricultural University, Beijing 100193, China. [2]These authors contributed equally: Jie Cao, Zhen Qin. ✉e-mail: mingmingxin@cau.edu.cn

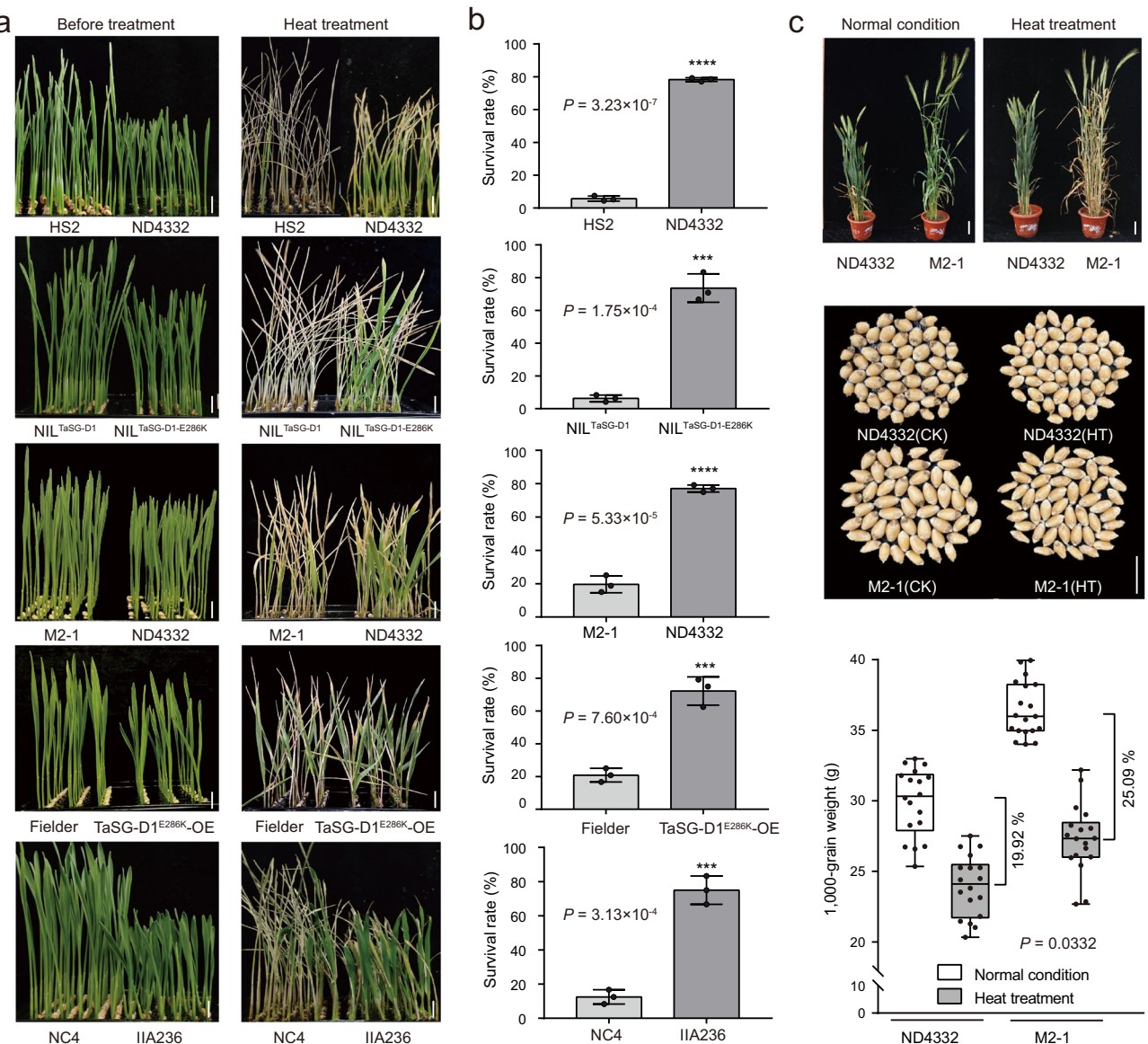

**Fig. 1 | *TaSG-D1^E286K* regulates heat tolerance in Indian dwarf wheat. a** Phenotypic analysis of heat responses of HS2 and ND4332, NIL^TaSG-D1 and NIL^TaSG-D1-E286K, M2-1 and ND4332, Fielder and *TaSG-D1^E286K-OE*, as well as NC4 and IIA236. Bar, 1 cm. **b** Statistical analysis of plant survival rates in response to heat stress (42 °C for 3 days). Statistical analysis was performed with GraphPad Prism 7 (v7.00). Each bar represents standard deviation. ****$P < 0.0001$, ***$P < 0.001$ ($n = 3$ biologically independent experiments; unpaired two-tailed $t$ test). **c** Phenotypic and statistical analysis of ND4332 and its mutant line M2-1 in response to heat stress at adult stage. Phenotype of representative plants are shown in upper panel, bar, 5 cm.

Performance of harvested grains from representative plants (50 for each plant) are shown in middle panel. CK, normal conditions; HT, heat treatment; bar, 1 cm. Statistical analysis of 1000-grain weight was shown in lower panel, performing with GraphPad Prism 7 (v7.00). The box plots display the interquartile range, comprising the first quartile, median, and third quartile, while the whiskers extend from the minimum to the maximum values. $P$ value represents the significant difference of yield loss in response to heat stress between two lines ($n = 18$ biologically independent samples; unpaired two-tailed $t$ test). Source data are provided as a Source Data file.

wsnp_Ex_c2258_4232538 and Xcau.3D-4 on the short arm of Chr.3D (Supplementary Fig. 1), which largely coincides with the genomic region responsible for semispherical grain in Indian dwarf wheat. Our previous study revealed that this seed morphological variation resulted from an amino acid substitution (E286K) in *TaSG-D1* (designated as *TaSG-D1^E286K*)[6]. We developed a near isogenic line of *TaSG-D1^E286K* (designated as NIL^TaSG-D1-E286K) and its near isogenic control (designated as NIL^TaSG-D1) by means of self-pollination of a residual heterozygous line from the F7 RIL population and marker-assisted selection. The genetic interval is ~11.3 Mb in length, and contains 84 high-confidence genes, among which, five genes process variations in coding sequence including the *TaSG-D1* (Supplementary Data 1). As expected, NIL^TaSG-D1-E286K exhibited enhanced heat tolerance compared to NIL^TaSG-D1 in terms of SSRH (73.6% vs. 4.2%, Fig. 1a, b).

To examine whether the *TaSG-D1^E286K* contributes to enhanced heat tolerance, we first examined the heat tolerance of additional five *T. sphaerococcum* with *TaSG-D1^E286K* allele and five *T. aestivum* accessions with *TaSG-D1* allele. All the Indian dwarf wheat lines exhibited higher seedling survival rates than common wheat in response to heat stress (Supplementary Fig. 2a, b). Next, we analyzed the performance of an EMS mutant line of ND4332 (M2-1) with loss-of-function of *TaSG-D1^E286K* in response to heat stress, which showed reduced heat tolerance compared to the wild type (19.6% vs. 77.1% of SSRH, Fig. 1a, b). By contrast, *TaSG-D1^E286K* overexpressors ('Fielder' background) showed better growth performance than the control under heat stress conditions (72.2% vs. 20.8% of SSRH, Fig. 1a, b), and its expression levels were positively correlated with the SSRH (Supplementary Fig. 3a). Moreover, we compared the heat tolerance between *TaSG-D1* OE line and

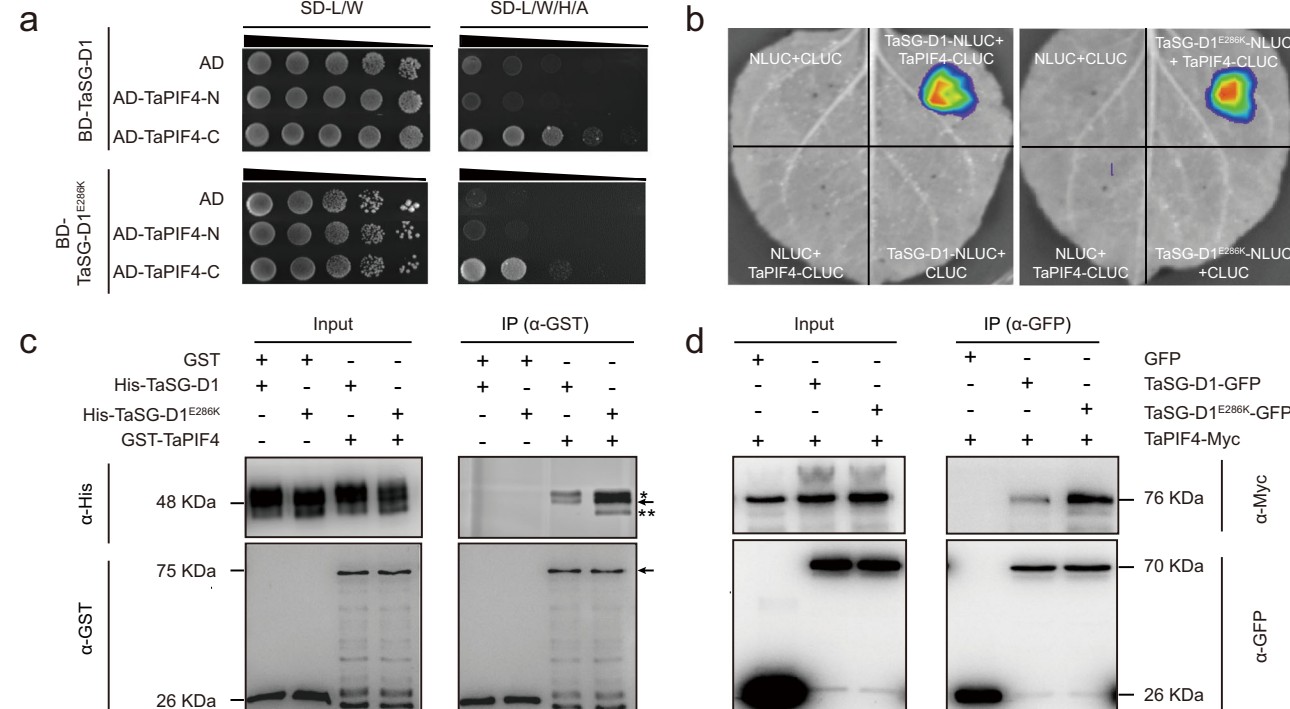

**Fig. 2 | TaSG-D1 and TaSG-D1$^{E286K}$ interact with TaPIF4. a** TaSG-D1/TaSG-D1$^{E286K}$ interact with the C-terminus of TaPIF4 in a Y2H assay. 35 mM 3-amino-1,2,4-triazole (3-AT) was used to suppress self-activation in the TaSG-D1$^{E286K}$ and TaPIF4 interaction assay. SD-L/W stands for media lacking Leucine and Tryptophan. SD-L/W/H/A stands for media lacking Leucine, Tryptophan, Histidine and Adenine. **b** TaSG-D1/TaSG-D1$^{E286K}$ interact with TaPIF4 in *N. benthamiana* leaves in an LCI assay. The experiment was performed five times. **c** TaSG-D1/TaSG-D1$^{E286K}$ interact with TaPIF4 in vitro in a pull-down assay. GST/GST-TaPIF4 were incubated with His-TaSG-D1/His-TaSG-D1$^{E286K}$. Protein mixture was immunoprecipitated with ProteinIso GST Resin and detected with anti-His (upper panel) and anti-GST (lower panel) antibodies. The arrow in lower panel (~75 KDa) indicates GST-TaPIF4. "**" in the upper panel indicates non-specific band, the arrow in the upper panel indicates non-phosphorylated His-TaSG-D1/His-TaSG-D1$^{E286K}$ (~48 KDa), "*" in the upper panel indicates self-phosphorylated His-TaSG-D1/His-TaSG-D1$^{E286K}$. Three independent experiments were performed. **d** TaSG-D1/TaSG-D1$^{E286K}$ interact with TaPIF4 in vivo in a Co-IP assay. 35 S: TaSG-D1-GFP/35 S: TaSG-D1$^{E286K}$-GFP/35 S: GFP and 35 S: TaPIF4-Myc were co-expressed in *N. benthamiana* leaves. Protein extract was immunoprecipitated with anti-GFP magnetic beads and detected using anti-GFP (lower panel) and anti-Myc (upper panel) antibodies. Three independent experiments were performed. Source data are provided as a Source Data file.

*TaSG-D1$^{E286K}$* OE line, which exhibited similar expression levels, and found that *TaSG-D1$^{E286K}$* OE line exhibited enhanced heat tolerance compared with *TaSG-D1* OE line (Supplementary Fig. 3b). In addition, IIA236, an EMS mutant line in NC4 background with a gain-of-function of *TaSG-A1* (E286K substitution in TraesCS3A02G136500, a homoeolog of *TaSG-D1* on chromosome 3A), exhibited improved heat tolerance compared to the wild type (75.0% vs. 12.5% of SSRH, Fig. 1a, b). We also examined the performance of near isogenic lines as well as ND4332 and its mutant line under heat stressed conditions at adult stage, and demonstrated that *TaSG-D1$^{E286K}$*-harboring lines exhibited enhanced heat tolerance compared with respective controls (Fig. 1c and Supplementary Fig. 4a, b). Moreover, field test found that ND4332 showed reduced yield loss compared with its mutant line M2-1 (19.92% vs. 25.09%, *P* = 0.0332), although wheat production is strongly affected in both lines by heat stress (Fig. 1c). Together, these results indicated that the E286K single amino acid substitution in TaSG-D1 accounts for the improved heat tolerance in *T. sphaerococcum*.

### *TaSG-D1$^{E286K}$* confers heat tolerance by regulating *TaPIF4*

To elucidate the regulatory mechanism of *TaSG-D1$^{E286K}$*/*TaSG-D1* in response to heat stress, we performed a yeast-two-hybrid (Y2H) assay to screen TaSG-D1$^{E286K}$/TaSG-D1-interacting proteins, and identified TaPIF4 (TraesCS5B02G380200) as a candidate partner (Supplementary Data 2). Through truncation analysis, we confirmed that the C-terminus of TaPIF4 is responsible for the interaction with TaSG-D1 and TaSG-D1$^{E286K}$ (Fig. 2a and Supplementary Fig. 4c). The physical interaction was further supported by luciferase complementation imaging (LCI), pull-down and co-immunoprecipitation assays (Fig. 2b, c, d).

To explore the biological role of *TaPIF4* in response to heat stress, we generated *TaPIF4* knockout mutants by simultaneously silencing its three homoeologs in the wheat cultivar 'Fielder' background via CRISPR-Cas9 genome editing. The triple knockout mutants exhibited impaired heat tolerance compared to the controls (72.9% vs. 25.0% vs. 27.08% for Fielder vs. *Tapif4-2* vs. *Tapif4-7* of SSRH, Fig. 3a and Supplementary Fig. 4d). In addition, *TaPIF4* overexpression lines exhibited improved heat tolerance compared to WT (31.25% vs. 68.75% vs. 70.83% for WT [KN199] vs. *TaPIF4-OE #1* vs. *TaPIF4-OE #2* of SSRH, Supplementary Fig. 5). To identify *TaPIF4*-dependent heat-responsive genes, we performed transcriptome sequencing of *TaPIF4* knockout line *Tapif4-7* (*Tapif4-KO*) and WT grown at normal conditions and heat stressed conditions (42 °C for 3 h and 6 h, respectively). In total, 2289, 4981 and 7278 genes were down-regulated in *Tapif4-KO* line compared with WT (log2[fold change] ≤ −1 and *FDR* < 0.05) at 0 h, 3 h and 6 h, respectively (Supplementary Data 3). Gene Ontology (GO) enrichment analysis revealed that those down-regulated genes at 0 h were mainly involved in photosynthesis, whereas those ones at 3 h and 6 h were predominantly enriched in "response to heat", "protein folding" and "response to reactive oxygen species" related GO categories (Supplementary Fig. 6a). Interestingly, *TaMBF1c* (TraesCS7B02G259000), which has been reported to positively confer heat tolerance via modulating protein folding related genes in wheat[7], was down-regulated in *Tapif4-KO* line compared with WT. Interestingly, *TaMBF1c* contained TaPIF4-binding motif in its promoter and the binding ability was

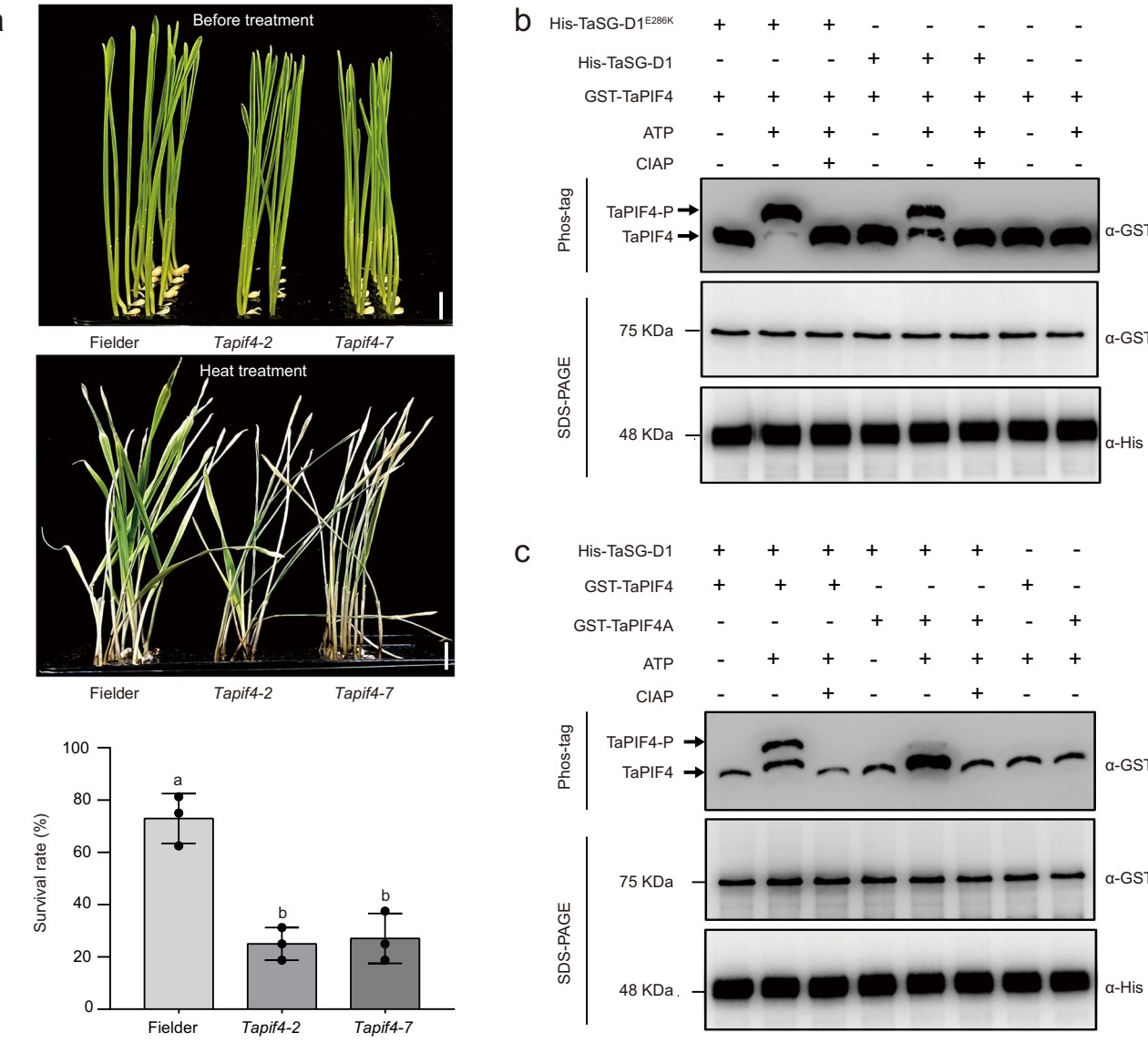

**Fig. 3 | Phosphorylation of TaPIF4 associates with heat tolerance in wheat.**
**a** Phenotypic and statistical analysis of Fielder and *TaPIF4* knockout mutants in response to heat stress (42 °C for 2 days). Scale bars = 1 cm. Statistical analysis was performed with GraphPad Prism 7 (v7.00). Error bars represent standard deviation, columns labeled with different alphabet were considered significantly different ($P < 0.05$, one-way ANOVA analysis-Tukey comparison; $n = 3$ biologically independent experiments). **b** TaSG-D1/TaSG-D1$^{E286K}$ phosphorylate TaPIF4 in vitro. The phosphorylation status of TaPIF4 was examined in a Phos-tag gel assay (top panel, anti-GST). TaPIF4-P and its arrow represent the phosphorylated band of TaPIF4. TaPIF4 and its arrow represent the unphosphorylated band of TaPIF4. CIAP: calf intestinal alkaline phosphatase. The protein abundances of TaPIF4 (middle panel,

anti-GST) and TaSG-D1/TaSG-D1$^{E286K}$ (bottom panel, anti-His) are shown in the SDS-PAGE gels. Three independent experiments were performed. **c** TaSG-D1 mediates the phosphorylation of TaPIF4 and its mutated form TaPIF4A in an in vitro phosphorylation assay. The phosphorylation levels of TaPIF4 and TaPIF4A were examined in a Phos-tag gel assay (top panel, anti-GST). TaPIF4-P and its arrow represent the phosphorylated band of TaPIF4. TaPIF4 and its arrow represent the unphosphorylated band of TaPIF4. CIAP: calf intestinal alkaline phosphatase. The protein abundances of TaPIF4/TaPIF4A (middle panel, anti-GST) and TaSG-D1 (bottom panel, anti-His) were monitored by SDS-PAGE. Three independent experiments were performed. Source data are provided as a Source Data file.

confirmed by Electrophoretic mobility shift assay (EMSA) (Supplementary Fig. 6b). Moreover, we identified 1299, 2809 and 3912 down-regulated genes with TaPIF4 binding motif (E-box motif) in their 1-kb promoter sequence at 0 h, 3 h and 6 h after heat stress according to the fimo software analysis ($P < 0.001$), which accounted for 56.7%, 56.4% and 53.8% of down-regulated genes, respectively, and GO enrichment analysis of these genes showed similar results to that of down-regulated genes (Supplementary Fig. 6c and Supplementary Data 3).

### TaSG-D1$^{E286K}$ enhances protein stabilization of TaPIF4

Since TaSG-D1$^{E286K}$ and TaSG-D1 are protein kinases, we performed in vitro phosphorylation assays to assess their ability to phosphorylate TaPIF4, and found that both proteins triggered TaPIF4

phosphorylation (Fig. 3b and Supplementary Fig. 7a, b). To identify the phosphorylation sites in TaPIF4, we conducted liquid chromatography-tandem mass spectrometry (LC-MS/MS) analysis. It was shown that Ser96, Ser97, Ser103, Ser133, Ser134, Thr135, Thr165, Tyr167, Ser174, Ser198, Ser199, Ser200, Ser414, and Thr429 in TaPIF4 protein are potentially phosphorylated (Supplementary Data 4). We then mutated these 14 sites (Ser/Thr/Tyr) to Ala to mimic the non-phosphorylated form of TaPIF4 (TaPIF4A). An in vitro phosphorylation assay revealed that the phosphorylation level was reduced in TaPIF4A compared to TaPIF4, indicating that these 14 amino acids were phosphorylated by TaSG-D1 (Fig. 3c and Supplementary Fig. 7c). To examine the potential function of TaSG-D1$^{E286K}$/TaSG-D1 mediated phosphorylation of TaPIF4, we also mutated the 14 phosphorylation

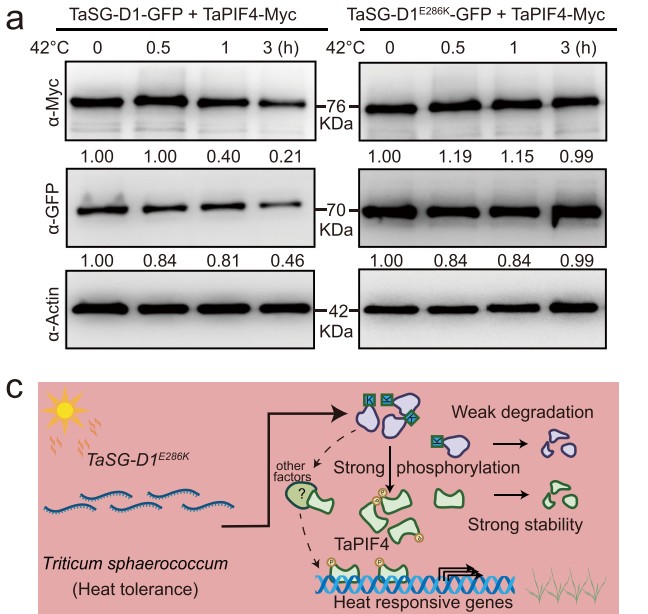

**Fig. 4 | TaSG-D1 and TaSG-D1$^{E286K}$ have different effects on TaPIF4 protein stability. a** TaSG-D1$^{E286K}$ enhances TaPIF4 stability compared to TaSG-D1 in response to heat stress. TaPIF4 was detected with anti-Myc antibody (top panel). TaSG-D1/TaSG-D1$^{E286K}$ was detected with anti-GFP antibody (middle panel). Actin was detected with anti-Actin antibody as a loading control (bottom panel). The relative abundances of TaPIF4 and TaSG-D1/TaSG-D1$^{E286K}$ were quantified using ImageJ software (v1.52i). The relative protein levels of TaPIF4/TaSG-D1/TaSG-D1$^{E286K}$ at 0 h were defined as 1.00. Three independent experiments were performed. **b** TaSG-D1$^{E286K}$ attenuates TaPIF4 degradation in a cell-free degradation assay. TaPIF4 abundance was examined with anti-GST antibody. TaSG-D1/TaSG-D1$^{E286K}$ were detected with anti-GFP antibody. Actin was detected with anti-Actin antibody as a

loading control. The relative protein abundance of TaPIF4 was quantified using ImageJ software. The relative protein level of TaPIF4 at 0 h was defined as 1.00. Three independent experiments were performed. **c** A working model for the role of the TaSG-D1/TaSG-D1$^{E286K}$-TaPIF4 signaling pathway in the response of wheat to heat stress (Created with BioRender.com). In hot areas of India and Pakistan, the protein of TaSG-D1$^{E286K}$ is stable, which then stabilizes TaPIF4 via strong phosphorylation and leads to improved heat tolerance of *T. sphaerococcum*. Under normal conditions, the protein of TaSG-D1 is unstable, which results in moderate degradation of TaPIF4 due to decreased phosphorylation, and causes reduced heat tolerance. Source data are provided as a Source Data file.

sites to Asp to mimic the phosphorylated form of TaPIF4 (TaPIF4D). TaPIF4D exhibited enhanced stability in response to heat stress compared to TaPIF4, and the E286K substitution did not significantly alter the stability of TaPIF4D. By contrast, TaPIF4A exhibited decreased protein stability compared to TaPIF4 under heat stress conditions (Supplementary Fig. 8). In addition, we found that TaPIF4D activated the *TaMBF1c* expression more efficiently than that of TaPIF4 (Supplementary Fig. 9a). To validate the biological significance of phosphorylated form of *TaPIF4*, we overexpressed *TaPIF4D* in wheat, and found that *TaPIF4D* overexpressors exhibited higher seedling survival rate than that of controls under heat stress conditions (29.17 % vs. 79.17 % vs. 81.25 % for WT [Fielder] vs. *TaPIF4D-OE-3* vs. *TaPIF4D-OE-9* of SSRH, Supplementary Fig. 9b). Moreover, we observed that phosphorylation-mimic form of *TaPIF4* (*TaPIF4D*) overexpression lines exhibited enhanced heat stress tolerance compared with *TaPIF4* overexpression lines, which showed comparable expression levels of *TaPIF4D* and *TaPIF4*, in terms of differential SSRH changes compared with their respective controls (37.50% and 39.58% increase for *TaPIF4* overexpression lines vs. 50.0% and 52.08% increase for *TaPIF4D* overexpression lines, Supplementary Fig. 5 and Supplementary Fig. 9b).

To further investigate how TaSG-D1$^{E286K}$/TaSG-D1 differentially regulate heat responses via the TaPIF4 pathway, we examined whether the E286K substitution influences the TaSG-D1$^{E286K}$/TaSG-D1 protein stability under normal and heat stress conditions. Interestingly, TaSG-D1 gradually degraded upon heat stress, whereas TaSG-D1$^{E286K}$ protein abundance exhibited a less degraded level (Fig. 4a). Moreover, compared to TaSG-D1, TaSG-D1$^{E286K}$ exhibited a stronger interaction with TaPIF4, as demonstrated in an LCI and pull-down assay (Supplementary Fig. 10a, b), resulting in increased phosphorylation level, and enhanced protein stability of

TaPIF4 in response to heat stress (Fig. 4a and Supplementary Fig. 10c). We then performed a cell-free degradation assay with various periods of time and demonstrated that the degradation rate of TaPIF4 was lower in TaSG-D1$^{E286K}$ protein solution than in TaSG-D1 under both normal and heat stressed conditions, and the TaPIF4D also exhibited increased stability (Fig. 4b and Supplementary Fig. 10d). Furthermore, we analyzed the TaSG-D1$^{E286K}$/TaSG-D1-associated TaPIF4 in comparison to the total amount of TaPIF4 in response to heat stress, and found that 23% and 88% of TaPIF4 protein were detectable after 3 h heat stress when co-transforming with TaSG-D1 and TaSG-D1$^{E286K}$, respectively, in *Nicotiana benthamiana* leaves, whereas only 12% of TaPIF4 protein remains stable in the control (Supplementary Fig. 11a). These results suggest that 11% and 76% of TaPIF4 are associated with and protected by TaSG-D1 and TaSG-D1$^{E286K}$, respectively, after 3 h heat treatment. In addition, the TaPIF4D protein stability was also improved in response to heat stress (Supplementary Fig. 11b). Together, these findings indicated that the increased protein stability and enhanced binding affinity of TaSG-D1$^{E286K}$ contribute to the reduced protein degradation of TaPIF4 by enhancing its phosphorylation level in response to heat stress, and then confers improved heat tolerance.

Collectively, we conclude the following: In *T. sphaerococcum*, TaSG-D1$^{E286K}$ protein exhibits weak degradation in response to heat stress due to E286K single amino acid substitution, and extensively interacts with, phosphorylates, and stabilizes TaPIF4, thereby improving the heat tolerance under heat stressed conditions; Whereas in *T. aestivum*, TaSG-D1 protein is unstable when subjected to heat stress and exhibits a weak interaction with TaPIF4, which leads to a decreased phosphorylation and an increased degradation level of TaPIF4 compared to TaSG-D1$^{E286K}$ in response to heat stress, resulting in reduced heat tolerance (Fig. 4c).

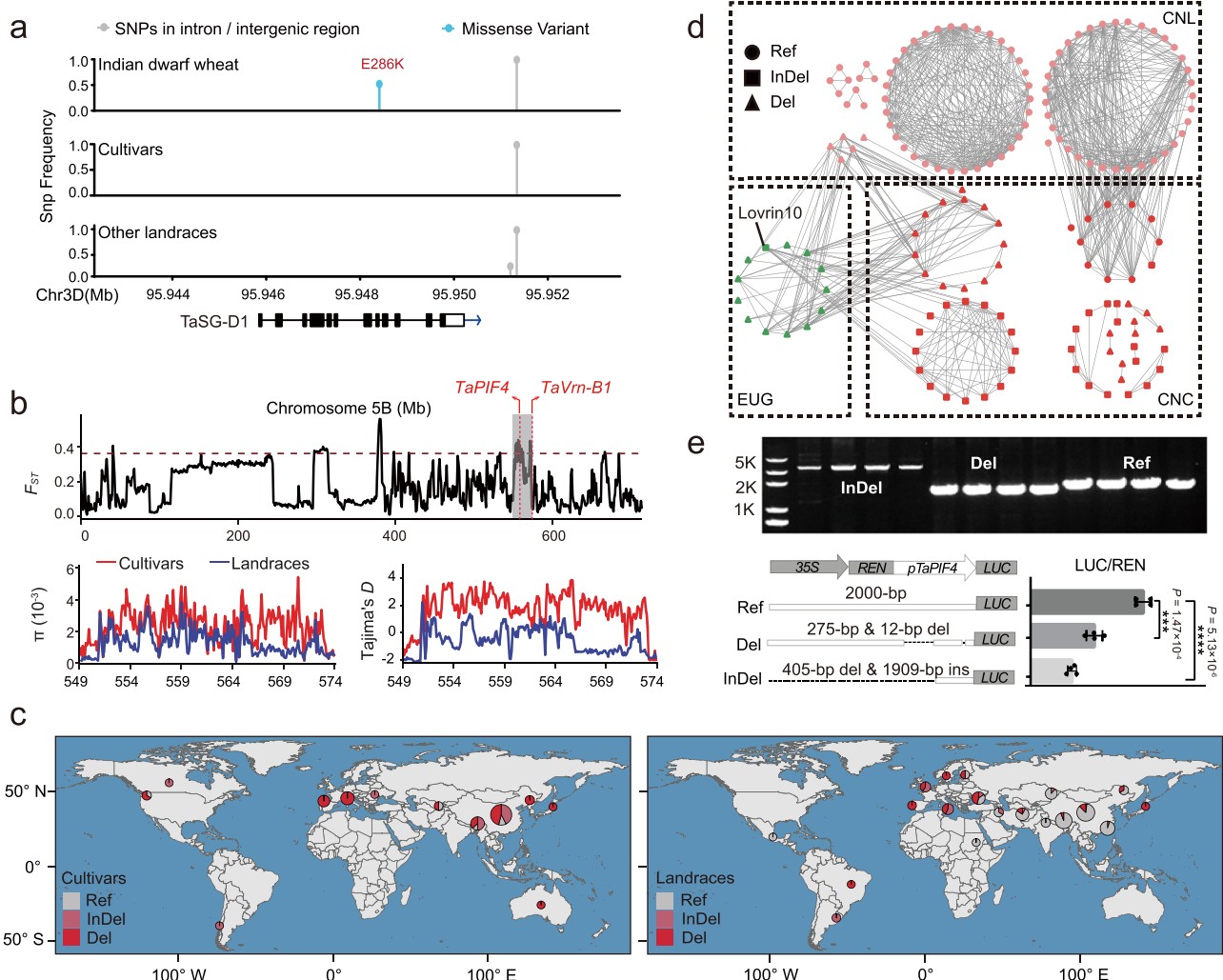

**Fig. 5 | *TaPIF4* shows artificial selection footprints in China during wheat selective breeding process. a** SNP variations of *TaSG-D1* in the global wheat accessions. SNP frequency was shown for *TaSG-D1* gene body and 2000-bp flanking sequence in Indian dwarf wheat, cultivars and other landraces besides Indian dwarf wheat, *n* = 331 wheat accessions. **b** $F_{ST}$ analysis identifies a 25-Mb genomic differentiation region between cultivars and landraces of China on chromosome 5B (upper panel). The gray shadow indicates the differentiation region (549.0–574.0 Mb) containing *TaPIF4* and *TaVrn-B1* gene. The red dotted horizontal line indicates the top 5% value (0.38); *π* and Tajima's *D* analysis further confirm the $F_{ST}$ result (lower panel), *n* = 125 and 116 wheat accessions. **c** Global distribution of three haplotypes of *TaPIF4* promoter. Gray (Ref) indicates the reference haplotype which is similar to IWGSC RefSeq v1.0 sequence; light red (InDel) indicates the haplotype with a 405 bp deletion and a 1909 bp insertion; red (Del) indicates the haplotype with 275 bp and 12 bp deletions, *n* = 331 wheat accessions. The map in (**c**) was created using geographic data from the Natural Earth project database. N: North latitude; S: South latitude; W: West longitude; E: East longitude. **d** 25-Mb

differentiation region-based germplasm network. Each node represents a wheat accession, only the edges with genetic similarity ≥20% are shown (159 of 331 wheat accessions). Light red indicates Chinese landraces (CNL); red indicates Chinese cultivars (CNC); Green indicates European germplasm (EUG). Node shapes indicate different haplotypes of *TaPIF4* promoter in (**c**). **e** Genomic variations of *TaPIF4* promoter influence its transcriptional level. PCR analysis confirms the sequence variations of *TaPIF4* promoter using different wheat accessions (upper panel). *TaPIF4* promoter activity analysis of three haplotypes in *Nicotiana benthamiana* indicates that sequence variations reduce *TaPIF4* expression levels in response to heat stress (42 °C for 3 h). Schematic diagram of the reporter constructs of three haplotypes: Ref, 2000-bp promoter fragment; Del, 1713-bp promoter fragment with 275-bp & 12-bp deletions; InDel, 337-bp promoter with 405-bp deletion & 1909-bp insertion. Statistical analysis was performed with GraphPad Prism 7 (v7.00). Each bar represents standard deviation. ****$P < 0.0001$, ***$P < 0.001$ (*n* = 4 biologically independent experiments; unpaired two-tailed *t* test). Source data are provided as a Source Data file.

## Genomic differentiation analysis of *TaSG-D1* and *TaPIF4*

To better understand the utilization of the $TaSG\text{-}D1^{E286K}$ allele during wheat breeding programs, we conducted an analysis of haplotype distribution among global wheat genetic resources according to published resequencing data[8–11] (Supplementary Data 5). Our findings indicate that the E286K variation is limited to *T. sphaerococcum* in India and Pakistan (Fig. 5a). In addition, we also investigated the variations in the 2-kb promoter region of *TaSG-D1* during wheat breeding history, as determined by analyzing published resequencing data of 159 modern cultivars and 172 landrace accessions[8–11], and revealed that the

promoter sequence is conserved and no genomic differentiation is observed between each other (Fig. 5a).

Next, we investigated sequence differentiation of the *TaPIF4* gene in global wheat resources, and found an artificial selection footprint during wheat breeding programs in China, where all wheat accessions possess *TaSG-D1* allele. $F_{ST}$ analysis identified a 25-Mb genomic differentiation region containing *TaPIF4* gene (549.0 Mb–574.0 Mb on chromosome 5B) between modern cultivars and local landraces of China (Fig. 5b). The observation was also supported by the results from nucleotide diversity (*π*) and Tajimas' *D* analysis (Fig. 5b). Further

investigation revealed that InDels (12-bp & 275-bp deletion; 1909-bp insertion & 405-bp deletion) preferentially occurred in the *TaPIF4* promoter of Chinese modern cultivars (88.80%) but not local landraces (9.48%, Fig. 5c and Supplementary Data 5). Genomic region-based germplasm network demonstrated that the 25-Mb selective fragment with InDels in the *TaPIF4* promoter was introduced into Chinese cultivars mainly from European germplasm but not inherited from local landraces (Fig. 5d). Consistently, Chinese modern cultivars have integrated extensive genetic variations from international germplasm[10], e.g., the elite wheat cultivar Lovrin10 was introduced into China from Romania in 1970's and was widely used as a founder parent due to its multiple disease resistance and high yield potential, which contributed greatly to the InDels expansion of *TaPIF4* promoter in Chinese cultivars (Fig. 5d).

To explore whether the natural variations associate with *TaPIF4* expression level in response to heat stress, we analyzed the 2-kb promoter sequence using PlantCare and identified 12 conserved stress-response elements (STRE, AGGGG). The 1909-bp insertion & 405-bp deletion interrupted six STREs, whereas the 12-bp & 275-bp deletions eliminated two STREs. We then performed a transient dual luciferase (dual-LUC) assay driven by the *TaPIF4* promoter, the *TaPIF4* promoter with 12-bp & 275-bp deletion and the partial *TaPIF4* promoter interrupted by insertion, respectively, in *Nicotiana benthamiana* leaves under heat stress conditions, and confirmed that the sequence variations led to dramatic decrease in LUC/REN activity in response to heat stress (Fig. 5e). This finding indicated that the artificially selective haplotypes of *TaPIF4* promoter are not favorable for heat tolerance improvement of modern cultivars in China. Interestingly, we found that the *TaVrn-B1* gene (TraesCS5B02G396600), a major target for selection of optimal seasonal flowering timing and grain production in wheat breeding[12–15], also locates in the 25-Mb divergent region and exhibits genomic differentiation between modern cultivars and local landraces (Fig. 5b and Supplementary Fig. 12). We identified two types of *TaVrn-B1* variation including a short deletion (~2500 bp) and a long deletion (~6800 bp) in the first intron. Interestingly, 87.5% wheat germplasm containing sequence deletions in *TaVrn-B1* also possess sequence variations in *TaPIF4* promoter (Supplementary Data 5). These lines of evidence suggest that the prevalence of *TaPIF4* promoter variations in Chinese cultivars is probably an incidental by-product during artificial selection of *TaVrn-B1* superior allele in modern wheat breeding process. However, we also noticed that a large portion of Chinese cultivars only contain sequence variations in *TaPIF4* but not in *TaVrn-B1*, this might be because breeders tend to explore *TaVrn-A1* variations to control flowering time during Chinese breeding history[14,16,17], whereas retention of sequence deletions in *TaPIF4* promoter indicates its involvement in regulating other agronomic traits, which merits further study.

## Discussion

To adapt to adverse environments, plants have evolved diverse regulatory mechanisms to rapidly respond the constraint, including the growth-defense trade-off regulatory mechanisms that maximize survival at the expense of growth fitness[18,19]. Our findings reveal that the E286K variation in TaSG-D1 is exclusive to *T. sphaerococcum* in India and Pakistan (Fig. 5a). This specific distribution is likely attributed to the high-temperature conditions experienced during the wheat growth period in the Indian subcontinent, where the introduction of the E286K variation is advantageous in overcoming this challenge. However, despite the enhanced heat tolerance, the E286K substitution in TaSG-D1 negatively impacts grain yield and has not been widely adopted following the Green Revolution. Whereas sequence variations in the promoter region of *TaPIF4*, a downstream target of TaSG-D1, are prevalent in wheat germplasm worldwide, yet the artificially selected haplotype of *TaPIF4* in modern Chinese cultivars has a negative impact on heat tolerance. This observation aligns with our previous study,

which revealed that the superior alleles for heat tolerance were eliminated during the modern wheat breeding process possibly due to their negative effects on yield potential[20]. Therefore, simultaneously optimization of agronomic traits and stress defenses via modulating growth-defense trade-off related genes remains a challenge because of their detrimental impact on crop growth.

Previous studies have demonstrated that increased accumulation of PIF4 contributes to enhanced hypocotyl elongation and triggers thermomorphogenesis in Arabidopsis under warm conditions by activating genes involved in the auxin pathway and cell wall organization[21–30]. Whereas in this study, our research uncovered a regulatory mechanism of heat stress tolerance involved in *TaPIF4* that the E286K variation in TaSG-D1, a negative regulator of BR, leads to increased accumulation of TaPIF4 protein, and improves heat stress tolerance by regulating a set of heat response genes including *HSPs, HSFs*, and *TaMBF1c* in wheat. But, besides *TaPIF4*, we propose that the involvement of unknown factors that act as downstream targets of TaSG-D1^E286K in the wheat heat stress response, because we found that stabilizing effects of TaSG-D1^E286K on the TaPIF4 protein are relatively mild as compared to the stabilizing effects of this mutation on TaSG-D1 itself. Moreover, near-isogenic lines of NIL^TaSG-D1E286K exhibited significantly enhanced heat tolerance compared to NIL^TaSG-D1, with survival rates of 73.6% and 4.2%, respectively. Whereas the survival rates for the wild type and *Tapif4* knockout mutants were 72.9% and 26.0%, respectively. Consistently, BIN2 has been reported to regulate multiple targets involved in the regulation of transcription, RNA processing and translation initiation etc.[31]. Of these, BRASSINAZOLE-RESISTANT1 has been revealed to bind to the promoter of *ERF49*, thereby regulate heat stress tolerance in Arabidopsis[32]. Moreover, ICE1 is also targeted and phosphorylated by BIN2, and is involved in the regulation of cold tolerance in Arabidopsis[33]. Hence, the *TaSG-D1*-mediated heat stress tolerance merits further study to comprehend the underlying network.

It is worthy noticing that BIN2, the homolog of TaSG-D1 in Arabidopsis (*Arabidopsis thaliana*), phosphorylates and destabilizes PIF4, whereas TaSG-D1 phosphorylates and stabilizes TaPIF4 protein in wheat. This discrepancy may be due to the different phosphorylation sites of PIF4 in the two species, because our study identified 14 phosphorylation sites of TaPIF4 through LC-MS/MS in wheat, which differs from the previously identified three phosphorylation sites of PIF4 in Arabidopsis[22]. Consistent with our hypothesis, it is reported that MPK3 can phosphorylate the Ser94 site of ICE1, leading to ICE1 degradation, and negatively regulating cold tolerance in Arabidopsis[34,35]; However, OsMPK3 can phosphorylate the Thr404, Thr406, Ser407, Thr412, and Ser433 sites of OsICE1 and enhances its protein stability, thereby positively regulating cold tolerance in rice (*Oryza sativa*)[36,37].

In addition, we observed that TaSG-D1^E286K mutation enhances TaPIF4 protein stability but reduces plant height in wheat. This finding appears to contradict the known biological function of *TaPIF4* that overexpression of *TaPIF4* results in increased plant height in wheat[38,39], whereas knockout of *TaPIF4* leads to a semi-dwarf phenotype as evidenced in this study. The discrepancy suggests that other TaSG-D1-dependent regulators are involved in the regulation of plant height in wheat. Consistent with our hypothesis, a recent study demonstrated that TaSG-D1^E286K phosphorylates and stabilizes DELLA protein Rht-B1b, the Green Revolution gene in wheat, and the Rht-B1b protein then interacts with TaPIF4 and inhibits its transcriptional activity during plant development, and finally reduces plant height in wheat[39]. Moreover, it has been reported that DELLA proteins interact with BZR1 and suppress its transcriptional activity, leading to growth inhibition in Arabidopsis[40]. Therefore, even though the stability of TaPIF4 protein is enhanced in TaSG-D1^E286K lines, its transcriptional activity is inhibited, resulting in decreased plant height.

Our study has identified a heat tolerance gene in wheat and elucidated its molecular network in response to heat stress. These findings not only enhance our understanding of heat tolerance

mechanisms in wheat but also provide potential gene resources to improve heat tolerance in wheat in the future.

## Methods

### Plant materials and growth conditions

The genetic linkage map of HS2/ND4332 RILs used in the present study was created by our previous study[6]. The line of HS2/ND4332, M2-1/ND4332, Fielder/TaSG-D1-OE/TaSG-D1[E286K]-OE, NC4/IIA236 and the five *T.sphaeroccum* were collected by our previous study[6]. The line of NIL[TaSG-D1]/NIL[TaSG-D1E286K] were constructed by means of self-pollination of a residual heterozygous line from the F7 RIL population and marker-assisted selection. We used the spring wheat cultivar 'Fielder' (*Triticum aestivum L.*) to carry out genetic transformations. The plants were grown in soil in a greenhouse with temperature of 25/20 °C (day/night) and 3000 lux light intensity (Philips). Growth chamber conditions at 22/20 °C (day/night) and 55% relative humidity were used to grow *N. benthamiana* plants.

### Heat tolerance evaluation and QTL analysis

We performed heat tolerance analysis of wheat seedling according to the previous study with minor modification[41]. The seeds were surface-sterilized with 1% sodium hypochlorite for 15 min and rinsed five times with distilled water. The seeds were stored in 4 °C under dark conditions for three days, and then transferred to room temperature for germinate. Germinated seeds were planted in a 96-well hydroponic box under 60% humidity, 22/22 °C temperature and 16 h/8 h period (day/night) conditions for 7 days. Water was refreshed daily. Then the 7-d-old seedlings were placed on hydroponic box with 1/4 Hoagland solution and treated at 42 °C for 2–5 days and recovered at 22 °C for 5–10 days. 1/4 Hoagland solution was changed every 2 days. The number of survival plants were recorded and used to calculate survival rates. For wheat heat tolerance assay at adult stage, we built a plastic tunnel to simulate heat stressed conditions. We used the temperature and humidity recorder (Elitech, RC-4) and the ElitechLog (v6.4.0) to record the temperature inside and outside the plastic tunnel every day. The heat treatment was applied for 34 days (May 5, 2021–June 7, 2021). Based on the genetic linkage map and the survival rates of heat tolerance phenotypic index at seedling stage, the QTL analysis was conducted using winQTLcart (v2.5) software by composite interval mapping (CIM).

### Yeast two-hybrid assay

The full length of *TaSG-D1/TaSG-D1[E286K]* coding sequences were cloned into pGBKT7 vector, the N terminal of TaPIF4 (TaPIF4-N, 1-274 aa) and the C terminal of TaPIF4 (TaPIF4-C, 275-447 aa) were cloned into pGADT7 vector. We transformed the different combinations of plasmids into yeast AH109. Synthetic complete medium lacking Leu and Trp (SD-L/W) was used to select transformed yeast cells, synthetic complete medium lacking Leu, Trp, His and Ade (SD-L/W/H/A) was used to detect interaction. The empty vectors pGADT7 and pGBKT7-*TaSG-D1/TaSG-D1[E286K]* were used as negative controls. Then we dropped 8 μL of serial decimal dilutions on the indicated medium plates and placed in a 28 °C incubator for 3 days. The candidate genes are listed in Supplementary Data 2. The primers used for plasmid construction are listed in Supplementary Data 6.

### Luciferase complementation imaging (LCI) assay

The coding sequences of *TaPIF4* was cloned into transient expression vector pCAMBIA1300-cLUC, and *TaSG-D1/TaSG-D1[E286K]* were cloned into pCAMBIA1300-nLUC. The resulting constructs were transformed into *Agrobacterium* strain GV3101. Different combinations of plasmids were co-infiltrated into 4-week-old *N. benthamiana* leaves. After 48–96 h, we used the Night SHADE LB 985 (Berthold Technologies, Bad Wildbad, Germany) system.to detect LUC activities of infiltrated *N.*

*benthamiana* leaves. The primers used for plasmid construction are listed in Supplementary Data 6.

### Recombinant protein purification

We constructed and purified His-TaSG-D1 (pET-28a)/His-TaSG-D1[E286K] (pET-28a)/His-TaSG-D1[E286K] (pET-32a)/GST-TaPIF4/GST-TaPIF4A as follows. We amplified and cloned *TaSG-D1/TaSG-D1[E286K]* cDNA into the pET-28a vector and pET-32a vector respectively, and *TaPIF4/TaPIF4A* cDNA were cloned into the pGEX-4T-2, then transformed into *E. coli* BL21 (DE3). At 37 °C, the cells were shaken at 200 rpm until the OD600 was 0.6. Then the cells were shaken for 12 h at 16 °C with 0.1 mM isopropyl-β-D-thiogalactopyranoside (IPTG) and purified using Proteinlso Ni-NTA Resin (TransGen Biotech, DP101-02)/Proteinlso GST Resin (TransGen Biotech, DP201-01). The primers used for plasmid construction are listed in Supplementary Data 6.

### Pull-down assay

For pull-down assays of interaction, 5 μg GST or GST-TaPIF4 coupled with Proteinlso GST Resin (TransGen Biotech, DP201-01) were incubated with 5 μg His-TaSG-D1 (pET-28a)/5 μg His-TaSG-D1[E286K] (pET-28a) at 4 °C for 2 h with gentle mixing. For pull-down assays of interaction strength, 5 μg GST or GST-TaPIF4 coupled with Proteinlso GST Resin (TransGen Biotech, DP201-01) were incubated with 2.5 μg His-TaSG-D1 (pET-28a) and 2.5 μg His-TaSG-D1[E286K] (pET-32a) at 4 °C for 2 h with gentle mixing. Then washed the mixture five times with washing buffer containing 50 mM Tris-HCl (pH 7.5), 50 mM NaCl, 0.1% Triton X-100 and 1 mM PMSF, then boiled for 5 min in 4×SDS loading buffer for immunoblot analysis. 10 μL sample was loaded for input immunoblot, and 15 μL sample was loaded for IP immunoblot. ProteinFind® Anti-GST Mouse Monoclonal Antibody (1:5000 dilution; TransGen Biotech, Catalog # HT601-01, Lot # Q20901), ProteinFind® Anti-His Mouse Monoclonal Antibody (1:5000 dilution; TransGen Biotech, Catalog # HT501-01, Lot # Q21105) and the secondary antibody of Goat Anti-Mouse IgG (H&L)-HRP Conjugated (1:5000 dilution; EASYBIO, Catalog # BE0102-100) were used to identify the proteins that were eluted from the resins.

### Co-immunoprecipitation assay

The coding sequences of *TaPIF4* was cloned into transient expression vector pCAMBIA1300 with a *Myc* tag at the C-terminus to generate *35 S: TaPIF4-Myc* construct, whereas the coding sequences of *TaSG-D1/TaSG-D1[E286K]* were introduced into the pCAMBIA1300-GFP vector to produce the *35 S: TaSG-D1-GFP* and *35 S: TaSG-D1[E286K]-GFP*. The above constructs were transformed into *Agrobacterium* strain GV3101 and co-infiltrated into *N. benthamiana* leaves according to different combinations. After 48–96 h, the total proteins of infiltrated *N. benthamiana* leaves were extracted by using lysis buffer (50 mM Tris−HCl at pH 7.5, 5 mM EDTA at pH 8.0, 150 mM NaCl, 0.1% TritonX-100, 0.2% NP-40, 0.6 mM PMSF, 20 μM MG132 and 1×protease inhibitor cocktail). After incubation in ice for 30 min, the samples were centrifuged at 13,000 *g* for 20 min. 20 μL Protein-A/G mix (Milipore, LSKMAGAG10) was added to the samples supernatants to incubate 1 h. Then 30 μL anti-GFP magnetic beads were added to the reaction with gentle rotation at 4 °C. After 2 h, we washed the beads using washing buffer containing 50 mM Tris−HCl (pH 7.5), 5 mM EDTA (pH 8.0), 150 mM NaCl, 0.2% NP-40, 0.1% TritonX-100 for five times. 10 μL sample was loaded for input immunoblot, and 15 μL sample was loaded for IP immunoblot. The immunoprecipitated proteins were detected using the primary antibodies of ProteinFind® Anti-GFP Mouse Monoclonal Antibody (1:5000 dilution; TransGen Biotech, Catalog # HT801-01, Lot # O21209) and ProteinFind® Anti-c-Myc Mouse Monoclonal Antibody (1:5000 dilution; TransGen Biotech, Catalog # HT101-01, Lot # P21018) and the secondary antibody of Goat Anti-Mouse IgG (H&L)-HRP Conjugated (1:5000; EASYBIO, Catalog # BE0102-100). The primers used for plasmid construction are listed in Supplementary Data 6.

## Generation of *TaPIF4* knockout and *TaPIF4D* overexpression wheat

*TaPIF4* knockout mutants were generated by CRISPR/Cas9 methods as follows: The sequences of the exons were used to design sgRNAs using the web-based E-CRISPR program (http://www.e-crisp.org/E-CRISP/). We synthesized two sgRNA sequences with *BsaI* cohesive ends. We amplified the MT1T2 vector using two pairs of primers containing the sgRNAs and cloned it into the CRISPR/Cas9 vector pBUE411. *TaPIF4D* overexpression lines were generated as follows: The coding sequences of *TaPIF4D* was cloned into pWMB110 vector to generate *pUBI*: *TaPIF4D-Myc* construct. The above constructs were transformed wheat cultivar 'Fielder' using *Agrobacterium tumefaciens* EHA105[42]. The sequences of sgRNAs and primers are listed in Supplementary Data 6.

## Electrophoretic mobility shift assay (EMSA)

In this study, oligonucleotide probes were synthesized and 5′ labeled with biotin. 5′ labeled oligonucleotide probes and GST or GST-TaPIF4 protein were incubated in 20 μL with 1×binding buffer (100 mM Tris, 500 mM KCl, 10 mM DTT; pH 7.5), 10% glycerol, 0.5 mM EDTA, 7.5 mM MgCl$_2$, 0.05% NP-40 and 50 ng μL$^{-1}$ poly (dI-dC) for 20 min at 25 °C. 5, 10, 20, 50 and 100-fold molar excess of an unlabeled DNA fragment as well as 100-fold mutated unlabeled DNA fragment was used in competition analysis to test the specificity to the DNA sequence. We separated binding mixtures by electrophoresis on 6% native polyacrylamide gels in 0.5×Tris borate EDTA buffer at 100 v for 40 min, and then transferred onto nylon membrane. LightShift™ Chemiluminescent EMSA Kit (Thermo Fisher Scientific, 20148) was used to detected the biotin-labeled probes. The primers are listed in Supplementary Data 6.

## RNA sequencing and data analysis

For RNA-seq analysis, 7-day-old wheat seedlings (Fielder and *Tapif4-7*) were treated with 42 °C (for 0 h, 3 h, 6 h) and were used to extract total RNA with TRIzol reagent (Invitrogen). Strand-specific RNA libraries were constructed and sequenced on a Novaseq 6000 platform by BerryGenomics Biotechnology (Beijing, China). Three biological replicates were performed for each sample. More than 70 million of 150-bp paired-end raw reads were generated for each replicate. The raw reads were processed with FASTP (v.0.19.4)[43] with parameters '−3 −5 -W 6 -l 30 -c'. The high-quality reads were then used for quantifying abundances of transcripts by using kallisto (v.0.45.0)[44]. The R package DEseq2 (v.1.24.0)[45] was used for differential expression analysis. Differentially expressed genes between conditions were identified according to |log2 (fold change)| ≥ 1 and adjusted $P < 0.05$ (Supplementary Data 3).

## Gene ontology enrichment analysis

Gene ontology (GO) enrichment analysis was conducted by using the ClusterProfiler (v.3.12.0)[46], all the annotated genes (107,891) in wheat reference genome were set as the background for hypergeometric test, and Benjamini-Hochberg method was used for multiple testing correction of *P* values. Finally, the top 10 significantly enriched GO terms of "cellular component", "molecular function and biological process" with an adjusted $P < 0.001$ were displayed in the heat map.

## Binding motif identification in *TaPIF4* promoter

E-box motif on the 1-kb promoter region of down-regulated genes in *Tapif4-7* compared with wild type was searched by using the fimo (v. 4.11.2) software[47], with the "--thresh 0.001" parameter, the background markov model was created from whole reference genome sequence by using 'fasta-get-markov' tools available in MEME suite[48] (Supplementary Data 3).

## Phos-tag mobility shift assay

Phos-tag SDS-PAGE was created in accordance with the manufacturer's instructions (Wako, 304-93521). On 10% SDS-PAGE gels containing 50 mM Phos-tag and 100 mM MnCl$_2$, total proteins from *N. benthamiana* leaves or the proteins of in vitro kinase assays were separated. The gel was transferred to a PVDF membrane after washing three times with transfer buffer (50 mM Tris, 40 mM Glycine) for 10 min each time, 10 mM EDTA was added in the first two times.

## In vitro phosphorylation assay

In vitro phosphorylation assays were performed as follows: GST-tagged TaPIF4 protein were co-incubated with His-tagged TaSG-D1/TaSG-D1$^{E286K}$ proteins in a total volume of 20 μL of kinase reaction buffer (25 mM Tris-HCl (pH 7.5), 12 mM MgCl$_2$, 5 mM ATP, and 1 mM DTT) at 37 °C for 1 h. The reactions were stopped by adding 4×SDS sample buffer. The reactions without ATP (Takara) and the reactions with CIAP (Takara) were used as negative controls. 10 μL sample was loaded for both immunoblot. The α-Casein (from Bovine Milk, Dephosphorylated, Wako, 038-23221) was used as positive controls and detected by CBB stain. The phosphorylation signal was detected by Phos-tag™ Acrylamide AAL-107 (Wako, 304-93521) according to the manufacturer's instruction. Anti-GST antibody (1:5000; TransGen Biotech, HT601-01, Q20901) and the secondary antibody of Goat Anti-Mouse IgG (H&L)-HRP Conjugated (1:5000, EASYBIO, BE0102-100) were used to detect the phosphorylation level of TaPIF4.

## In vivo phosphorylation assay

The constructs of *35 S: TaPIF4-Myc* and *35 S: TaSG-D1-GFP/35 S: TaSG-D1$^{E286K}$-GFP* were transformed into *Agrobacterium tumefaciens* strain GV3101 separately. Various blends of plasmids were co-expressed into 4-week-old *N. benthamiana* leaves. After 48–96 h, the infiltrated *N. benthamiana* leaves were treated under 42 °C for 3 h. Next, total proteins were extracted using 4× protein loading buffer and boiled for 10 min, and 10 μL sample was loaded for immunoblot. The phosphorylation signal was detected by Phos-tag™ Acrylamide AAL-107 (Wako, 304-93521) according to the manufacturer's instruction. Phosphorylation of TaPIF4 were detected using anti-Myc antibody (1:5000; TransGen Biotech, HT101-01, P21018).

## Liquid chromatography-tandem mass spectrometry

We performed liquid chromatography-tandem mass spectrometry (LC-MS/MS) assay to identify putative phosphorylation sites of TaPIF4 as follows: GST-tagged TaPIF4 (100 ug) protein were co-incubated with His-tagged TaSG-D1 (50 μg) proteins in a kinase reaction buffer (25 mM Tris-HCl [pH 7.5], 12 mM MgCl$_2$, 5 mM ATP, and 1 mM DTT) at 37 °C for 1 h. The total reaction proteins in vitro phosphorylation assays were reduced by DTT, alkylated by IAM, digested overnight at 25 °C by trypsin and then diluted by 0.1% [v/v] formic acid. After centrifuge at 12,000 *g* for 20 min, the collected supernatant was analyzed by nano-Acquity nano HPLC (Waters, Milford, MA, USA) coupled with a Thermo Q-Exactive high resolution mass spectrometer (Thermo Scientific, Waltham, MA, USA) for LC-MS/MS assays. Then, the MASCOT search engine (Matrix Science, Mascot 2.6.0) was used to analyze the raw MS/MS data. The UniProt wheat sequence database was used to search peptides. The search parameters were employed as follows: trypsin was set as a specific enzyme, a maximum missed cleavages were set to 2, fixed modification contain carbamidomethylation (Cys) and variable modification contain oxidation (Met) and phosphorylation (Ser, Thr and Tyr). Peptide and fragment tolerances were 10 ppm and 0.02 Da. Threshold value of positive peptides was MASCOT score >35 corresponding to false positive rate <5%. The positive peptides and identified phosphorylation sites were listed in Supplementary Data 4.

## TaPIF4 stability assay

The constructs *35 S: TaPIF4-Myc*, *35 S: TaPIF4A-Myc*, *35 S: TaPIF4D-Myc* and *35 S: TaSG-D1-GFP/35 S: TaSG-D1^E286K-GFP* were transformed into *Agrobacterium tumefaciens* strain GV3101 separately. Various blends of plasmids were co-expressed into 4-week-old *N. benthamiana* leaves. Between 48 and 96 h after transfection, the total proteins were extracted for immunoblotting. For the TaPIF4 and TaSG-D1/TaSG-D1^E286K stability in different temperature gradients, the infiltrated *N. benthamiana* leaves were treated under 42 °C for 0 h, 0.5 h, 1 h, 3 h. Next, total proteins were extracted for immunoblotting. 10 μL sample was loaded for both immunoblot. The proteins were detected using anti-Myc antibody (1:5000; TransGen Biotech, HT101-01, P21018), anti-GFP antibody (1:5000; TransGen Biotech, HT801-01, O21209), anti β-Actin Mouse Monoclonal antibody (1:5000 dilution; CWBIO, Catalog # CW0264M, Lot # 01265/35721, Clone # 6D1) and the secondary antibody of Goat Anti-Mouse IgG (H&L)-HRP Conjugated (1:5000, EASY-BIO, BE0102-100). The primers used for plasmid construction are listed in Supplementary Data 6.

## Cell-free protein degradation assay

To investigate the degradation of TaPIF4 in vitro, we carried out cell-free protein degradation experiment. Briefly, the GV3101 contained *35 S: TaSG-D1-GFP/35 S: TaSG-D1^E286K-GFP* were co-expressed into 4-week-old *N. benthamiana* leaves. After 48–96 h, total proteins were extracted in buffer containing 50 mM Tris- HCl (pH 8.0), 500 mM sucrose, 1 mM MgCl$_2$, 10 mM EDTA (pH 8.0), and 1 mM DTT. Then the extracted total proteins incubate with purified GST-TaPIF4 protein with 10 mM ATP at 30 °C (normal condition) and 42 °C (heat stressed condition) in different time gradients. After incubation, 10 μL sample was loaded for both immunoblot, and anti-GST antibody (1:5000; TransGen Biotech, HT601-01, Q20901) was used to detect the TaPIF4 protein.

## Transient dual luciferase (dual-LUC) assay

To generate *pTaPIF4Ref: LUC* and *pTaPIF4Del: LUC* and *pTaPIF4InDel: LUC* constructs, we amplified 2000-bp, 1713-bp and 337-bp promoter fragment of *TaPIF4* from wheat varieties of Baimangxiaomai, AK58 and Nongda5181, respectively. Then these sequences were inserted into pGreenII 0800-*LUC* as reporter vectors and were transformed into *Agrobacterium* strain GV3101. Next, the vectors were infiltrated into 4-week-old *N. benthamiana* leaves, which were subjected to heat stress at 42 °C for 3 h. By Dual-Luciferase Reporter Assay system (Promega, E1960), we quantified the activity of firefly luciferase under control of *TaPIF4* promoters and Renilla luciferase under control of 35 S promoter with a multimode reader Spark (TECAN) and the software SparkControl (v2.1). Data are presented as the ratio of luminescent signal intensity for reporter vs. internal control reporter (*35 S: REN*).

For the dual-LUC assay testing transcriptional activity of TaPIF4 and TaPIF4D, the constructs of pGreenII 0800-*pTaMBF1c: LUC* and pCAMBIA1300- *35 S: TaPIF4-Myc* and pCAMBIA1300- *35 S: TaPIF4D-Myc* were generated. The construct of pCAMBIA1300- *35 S: Myc* was used as control. Relative LUC activity (LUC/REN) was calculated to reflect transcriptional activity. The primers used for above constructions are listed in Supplementary Data 6.

## Genomic variation calling

We collected 331 sets of previous published resequencing data of hexaploid wheat accessions from NCBI Sequence Read Archive database[8–11], and divided them into six groups according to their geographical location, that is, CNL, CNC, EUL, EUC, EXL, EXC (CN, Chinese; EU, European; EX, Others; L, landraces; C, cultivars). Sequence variation calling was performed as follows: we firstly mapped the clean reads onto 'Chinese Spring' reference genome (IWGSC RefSeq v1.0)[49] using BWA-MEM. After removing low

mapping quality and PCR duplicate reads (Bamtools v2.4.1 and Samtools v1.3.1)[50,51], HaplotypeCaller, GenotypeGVCFs and VariantFiltration modules of GATK v3.8[52] were employed for variation identification, joint calling and variation filtering, respectively. VariantFiltration function with the parameter "QD < 2.0, FS > 200.0," and "ReadPosRankSum < −20.0||DP > 30||DP < 3" was used for preliminary InDels filtering, and InDels with MAF ≥ 0.05, missing rate ≤40% or bi-allelic sites for further removed. Next, we calculated reads depth of 2000-bp sequence upstream from the translation start site of *TaPIF4* to identify large InDels. Then, we performed PCR analysis followed by sequencing using 12 accessions to verify our genotyping result. The information for 331 wheat accessions is listed in Supplementary Data 5. The primers used for PCR are listed in Supplementary Data 6.

## Genomic region detection responsive to artificial selection

To detect differentiation regions during selective breeding process in China, we defined the 116 and 125 samples as landraces and cultivars according to published variety information. The differentiation index ($F_{ST}$) was computed utilizing VCFtools (v0.1.13)[53] with a window size and step size of 100 kb, with top 5% quantiles used for differentiated region identification. Nucleotide diversity $\pi$ and Tajima's $D$ statistics were also performed using VCFtools (v0.1.13) to estimate sequence variation within sample groups.

## Reporting summary

Further information on research design is available in the Nature Portfolio Reporting Summary linked to this article.

## Data availability

The RNA-seq data have been deposited in the NCBI database under accession PRJNA906524. Previously published resequencing data[8–11] were used in this study; they are available in Sequence Read Archive under accession PRJNA476679, PRJNA596843, PRJNA439156, PRJNA663409, PRJNA597250. In addition, the raw sequence data of previously published re-sequenced accessions[9,10] used in this study are available in the Genome Sequence Archive under accession number CRA001870 and CRA001951. The Chinese Spring wheat reference genome (IWGSC RefSeq v1.0) is publicly available at [https://wheat-urgi.versailles.inra.fr/Seq-Repository/Assemblies]. The wheat protein sequence database is publicly available at UniProt [https://www.uniprot.org/taxonomy/4565]. Source data are provided with this paper.

## Code availability

The scripts used in the study have been deposited at GitHub [https://github.com/QinZhen1995/CAU-TaSG][54].

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

## Acknowledgements

We would like to thank X. Y. Kong (Chinese Academy of Agricultural Sciences) for kindly providing seeds of *TaPIL1*/*TaPIF4* overexpressing transgenic lines. This work was supported by the National Key Research and Development Program (SQ2022YFD1200002) to Y.Y, the National Natural Science Foundation of China (31922067) to M.X. Sci-Tech Special project of Inner Mongolia (NMKJXM202201) to M.X. and the Chinese Universities Scientific Fund (2021TC036) to M.X.

## Author contributions

M.X. and Q.S. conceived the project. J.C., Z.Q., G.C. and Z.C. collected plant materials. J.C. and G.C. performed the research. Z.Q., J.C., X.C., W.G., H.P., Y.Y., Z.H. and Z.N. analyzed the data. M.X. and Q.S. wrote the manuscript.

## Competing interests

The authors declare no competing interests.
