## [Peer Review File · Nature Communications]

Natural variation of STKc_GSK3 kinase TaSG-D1 contributes to heat stress tolerance in Indian dwarf wheatReviewers' Comments:

Reviewer #1:

Remarks to the Author:

The paper titled "Natural sequence variation of STKc_GSK3 kinase TaSG-D1 contributes to thermotolerance in Indian dwarf wheat" presents a study that identified that the mutation E286K on the gene *Tasg-D1* is responsible for the protein TaSG-D1's role in thermotolerance.

The authors subsequently showed both WT (TaSG-D1) and the (amino acid substitution E286K), which the authors confusingly named *Tasg-D1*, suggesting this is a recessive allele - is it? unless I am missing something here as there is significant amount of genetic work in this study, including QTL). The authors then demonstrated that both TaSG-D1 and *Tasg-D1* (here-forth E286K mutant to minimise confusion in this report) interacts with C-terminal of PIF4 through Yeast-2-hybrid, Co-PI and LCI assay in *N. benthamiana*. Additionally, this study demonstrated PIF4's role in thermotolerance through CRISPR-KO of wheat followed by transcriptomic studies on PIF4 mutants at higher temperatures - It is well known that that PIF4 is an important factor in plant thermotolerance. Although most work is limited to arabidopsis and it is nice see this confirmed in a monocot.

The authors next demonstrated that both *tasgD1* and *taSG-D1* were found to trigger PIF4 phosphorylation. However, it is difficult to assess whether this is really case as I can not see any obvious differences (Figure 3B and C). We next move to the most interesting revelation where the authors revealed that mutation E286K is responsible for SG-D1 protein stability during heat stress at 42°C. The study ends with PIF4 loci analysis and the relationship of its mutations and geographical distribution.

Overall, I think the main finding of this work important, that TaSG-D1 interacts with PIF4 and that mutation E286K contributes to the TaSG-D1 protein stability. However, the results at its current state is preliminary and is more appropriate for a specialised journal rather than Nature communications. Below are my major and minor comments

Major comments.

1. Naming E286K mutant of SG-D1 to *sg-D1* makes the whole manuscript extremely confusing to read. The authors should at least consider SG-D1-E286K.
2. It is nice that the author demonstrate both WT and E286K mutant interacts with PIF4 and that the The authors tried to However, I find the claim that TaSG-D1 mediates phosphorylation of TaPIF4 unconvincing. Positive and negative controls are essential for this study (e.g. Figure 3B and C) - essentially all bands look the same to me whether in presence of ATP, either TaSG-D1, etc...
3. While I agree the mutation E286K do contribute to thermo-stability of TaSG-D1 protein. the results suggested the lower level of PIF4 associated (whether wt or mutant) simply correlates with protein abundance (i.e. less SG = less associated PIF4). It would be interesting for the authors to assess the total amount of PIF4 in comparison to SG-associated PIF4.
4. Kinase activity for *tasgD1* and *taSG-D1* -on PIF4 phosphorylation where are the controls? everything look the same to me.
 - Can use a recombinant protein, phosphatase treated as control.
 - some sort of quantification to transform the band intensity to bar plots (e.g. phos-tag vs anti-GST).
5. Major rewrite of this manuscript is crucial. Currently it is rather eclectic, especially at most of the work is really on TaSG-D1 and its effect on PIF4, and therefore thermotolerance e.g. discuss TaSG-D1 distribution alongside PIF4 instead of PIF4 alone - while I appreciate the amount of work required to generate a wheat *pif4* mutant, this paper is not about PIF4 alone.

Minor comment

please don't put band quantification on top of bands. Either include the values under the bands or plot it next to the gel images

Reviewer #2:

Remarks to the Author:

Cao et al. identify in this work a Tasg-D1 thermotolerance gene in the Indian dwarf wheat (*Triticum sphaerococcum*), encoding a STKc_GSK3 kinase with an E286K substitution as compared to the TaSG-D1 wild type. This amino acid substitution stabilizes in heat stress conditions the Tasg-D1 protein and confers increased binding affinity towards its TaPIF4 target, causing elevated TaPIF4 phosphorylation levels and greater stability of the protein, which correlates with an enhanced thermotolerance of dwarf wheat. In line with this regulation, loss of function of Tasg-D1 or TaPIF4 impair wheat thermotolerance. Comparative genomic studies identified the presence of several InDels in the TaPIF4 promoter of modern Chinese wheat cultivars, leading to decreased TaPIF4 expression levels in response to heat stress. Variation in this region appears to have been introduced from European germplasm, while it is absent in local landraces. Those were likely co-retained during selection of the TaVrn-B1 superior allele, these findings hence uncovering TaPIF4 as a potential locus for wheat thermotolerance improvement.

Via QTL mapping of a *T. sphaerococcum* and *T. aestivum* RIL population, authors had earlier identified a *sphaerococcum* allele which carries a E286K substitution in the TaSG-D1 gene, a GSK3 kinase homologous to Arabidopsis BIN2, as conferring the semispherical grain morphology of Indian dwarf wheat. In this work authors show that this substitution is also responsible for enhanced thermotolerance of Indian dwarf wheat. NILTasg-D1 lines with this allele display a higher survival to 3 days 42°C treatments than near isogenic NILTaSG-D1 lines. Authors show that heat tolerance is suppressed in *sphaerococcum* M2-1 mutants, with a loss-of-function mutation in the Tasg-D1 gene, while a better heat stress growth performance is observed in Tasg-D1 OE lines. These findings strongly support the E286K substitution to account for the improved thermotolerance trait. A Y2H screen for Tasg-D1 interactors identified the wheat TaPIF4 factor, which is confirmed by Luciferase complementation, pull-down and co-IP assays. CRISPR-Cas9 knock-out mutations of TaPIF4 in the Fielder cultivar are here shown to lead to impaired thermotolerance, and to correlate with down-regulated expression of several "heat responsive", "protein folding" and "ROS responsive" transcripts under heat stress conditions. Both Tasg-D1 and TaSG1-D1 were found to phosphorylate TaPIF4, allowing LC-MS/MS identification of 14 potential phosphorylation sites. Still, mutation of these residues reduces TaPIF4A phosphorylation, but did not impair in vitro phosphorylation of this protein by Tasg-D1. Phosphomimic Ser/Thr to Asp substitutions within these residues are however shown to enhance TaPIF4D heat stability, while this to be reduced in the TaPIF4A protein. Moreover, TaSG-D1 was observed to be progressively degraded under heat-stress conditions, while Tasg-D1 abundance remain unaffected. TaPIF4 is as well shown in cell-free degradation assays to be more stable in the presence of Tasg-D1 than TaSG-D1, suggesting Tasg-D1 stabilization contributes to higher TaPIF4 phosphorylation and to the accumulation of the protein under heat stress conditions, which correlates with improved thermotolerance.

The Tasg-D1 mutation negatively affects grain yield and was not introduced in Green Revolution cultivars. Remarkably, comparative genomics of modern and local China landraces identified a 25-Mb hypervariable footprint region around the TaPIF4 locus. Various InDels interrupting putative stress-response were indeed observed in the 2-kb TaPIF4 promoter region, which are absent in local landraces. Expression studies showed these changes reduce heat activation and thus may play a role in lowering thermotolerance of modern cultivars. The TaVrn-B1 gene locates in the 25-Mb footprint region, indicating that this region was co-introduced together with the TaVrn-B1 superior allele during modern wheat breeding. Modification of the TaPIF4 promoter is thus proposed as a target for improving wheat thermotolerance.

Overall, this is an exciting piece of work that provides solid evidence for the E286K substitution in the STKc_GSK3 *T. sphaerococcum* allele to confer enhanced tolerance to heat stress. Notably, this substitution locates into the same TREE domain as the Arabidopsis bin2.1 mutation described in Chory's lab (Li et al., 20021) and the ucu1 mutant identified in Micol's laboratory (Perez-Perez et al., 2002), and shown to encode a hypermorphic allele. Mutations in the TREE domain prevent BR-

mediated BIN2 inhibition (Peng and Li, 2003) and were established to result in increased BIN2 stability (Vert and Chory 2006; Peng et al., 2008) as it is here shown for the wheat Tasg-D1 protein. Arabidopsis BIN2 is on the other side reported to phosphorylate and destabilize the Arabidopsis PIF4 and PIF3 transcription factors (Bernardo-García et al., 2014; Ling et al., 2017). Therefore, in opposite to the results reported here for wheat, phosphorylation was found in Arabidopsis to lead to the destabilization of both proteins. Such contrasting results might be indicative of a different behavior of the wheat PIF factors, or most likely that stability of these proteins is reversed under heat stress (42°C). Indeed, ND42332 and NILTasg-D1 lines show a clear dwarf phenotype suggestive of a reduced stability of TaPIF4 under control conditions. Surprisingly, such dwarfing phenotype is less evident in Tasg-D1 OE lines. Additionally, a semidwarf phenotype is observed in Tapif4-2 knock-out mutants and to a lesser extent in the Tapif4-7 lines, again supporting a role of TaPIF4 in promoting cell elongation in a similar manner as shown in Arabidopsis (why do these lines display a different phenotype when both correspond to ko mutants?). As such, it would be critical to carry out similar cell-free degradation assays as shown in Figure 4, under control temperatures, and thus establish that BIN2-dependent control of PIFs stability is opposite under heat stress and control conditions, or rather than the wheat and Arabidopsis factors display an opposite response. Although the findings nicely imply a role of TaPIF4, downstream of Tasg-D1, in conferring thermotolerance, they do not exclude the participation of other transcriptional factors in conferring this trait. AtPIF4 was shown to heterodimerize with BES1/BZR1, and this complex to play a role in regulating BR-synthesis (Martínez et al., 2018). Authors should discuss this possibility and more importantly discuss their findings in the context of the previous knowledge generated in Arabidopsis.

Reviewer #3:

Remarks to the Author:

The authors present a study looking at natural sequence variations in a protein kinase that has ramifications for thermotolerance in wheat. The authors present a number of lines of evidence to suggest a mechanism by which a mutation in the protein kinase results in weaker degradation of PIF4 leading to thermotolerance.

While the authors present a reasonable story, there are substantial issues that need to be rectified.

Major issues:

1. Many of the figures are of substandard quality and need major attention.
 - a. multiple bar graphs depicting data without the presentation of replicates as is standard in publication. The box plot of Figure 1 is a good example that should be replicated across the manuscript.
 - b. Figure 4 and S6 have blot panels that are cropped too tightly with densitometry values overlaid atop the blot. The numerical values should be placed where they do not obstruct the blot. Further, each of the presented blot windows need to be widened.
 - c. Figure S6c should have all panels the same size with corresponding molecular markers to be able to properly assess the size shift claimed. Further, from the blot presented it is unclear if there is in fact a distinct size shift as there is a large smear, unlike what is presented in Figure 3.
 - d. Figure 3 should be corroborated by an anti-phospho blot as it is not immediately clear if there is a size shift or if the blot ran awkwardly. PhosTag gels have a tendency to run awkwardly, so a blot of these samples would confirm the result.
 - e. Figure 2: It is unclear which band is PIF4 or if these are degradation products or even non-specific binding of the antibody.
 - f. Full length protein stained gels and corresponding western blots should be provided in a supplemental figure to allow the reader to assess the purity of the preparations used in the assays.
 - g. Figure s5 the LCMSMS data is unreadable. This data should be assembled into a supplemental table that aligns with field standards for this data type with the corresponding data common to PTM site identification included (e.g. site localization reliability value, score, amongst multiple other data

types).

h. Figure s4. It is unclear what background was used to perform the GO enrichment analysis and what the statistical threshold was and the correction used was.

i. All figures containing western blots do not provide any indication of replication nor how much sample was loaded per lane.

j. A threshold for differential gene expression of $FDR < 0.05$ and fold change of < 0.5 is too loose. The FDR p-value is fine, but minimally a FC of < 1.5 is required to minimally align with field standards.

k. Pictures of replicates for the LUC assay should be shown in a supplemental figure.

2. Multiple methods are inadequately described.

a. For the GO enrichment what was the FDR correction used? See other GO enrichment commentary above in Major Concern 1.

b. The LCMSMS methods are completely deficient. So much so, that it is too much to fully comment on. The authors need to consult other leading manuscripts in the field describing these methods and carefully describe what they did here. The current approach of citing another paper is inappropriate for methods this complex.

While the results are intriguing, the aforementioned deficiencies in presentation, methods and results require substantial attention.

Reviewer #4:

Remarks to the Author:

Heat stress is one of the major threats for agriculture, especially under the condition of global warming. Dissection of the genetic basis of thermotolerance helps to breed heat resistant crops. However, not much progress has been made. In the manuscript by Cao et al., the author presented the cloning of a thermotolerance QTL gene, TaSG-D1. An amino acid substitution (E286K) in TaSG-D1 resulted in increased stability of the protein and stronger interaction with its target, TaPIF4, which then regulates targets like TaMBF1c to enhance heat tolerance. Interestingly, they found that several weak alleles of TaPIF4 (variations in promoter), instead of the wild type allele are widely spread in Chinese modern wheat cultivars and they proposed that this phenomenon is caused by the linkage with TaVrn-B1. Breaking this linkage may bring new opportunity to enhance heat resistance in wheat. As far as I know, this is the first report about cloning of heat resistant gene by forward genetic approach. I have several concerns before I could recommend it for publication.

Major:

1. The authors provided several lines of evidence like phenotype of OE lines and EMS mutant to approve the candidacy of TaSG-D1. However, in the concept of forward cloning, they are still indirect evidence. Complementary test or functional validation of both alleles is necessary to demonstrate that the phenotype variation of the NIL is caused by the candidate gene. Meanwhile, more information of the interval (size, genes existed) should be provided.

2. One interesting finding is that Tasg-D1 has a strong binding ability to TaPIF4 than TaSG-D1. However, to say that Tasg-D1 interacts more strongly with TaPIF4, only one LCI assay is insufficient, especially when there is no reference to perform calibration (Figure S6 B). Would it be possible to perform a competing interaction assay? For instance, equal amount of differentially tagged Tasg-D1 and TaSG-D1 could be co-incubated with TaPIF4 in one tube for later pull-down.

3. Tasg-D1 and TaPIF4 are working together to confer heat tolerance but both TaPIF4(knock out line) and TasgD1(functional allele) showed a semi-dwarf phenotype. Any explanation for this?

4. The time series RNA-seq could provide more information but the data analysis is a bit too weak. At least, the author could calculate how many DEGs contain PIF4 binding motif. Validation for binding of TaPIF4 to TaMBF1c (by EMSA or other means) is required.

5. Due to negative effect on seed size, Tasg-D1 has not been widely used. I am wondering whether there are other types of variations than E286K on TaSG-D1 gene existing in the germplasm, particularly in promoter? By the way, is expression level correlated with the performance of Tasg-D1

(maybe more Tasg-D1 OE lines) in response to heat stress?

6. Tasg-D1 phosphonates TaPIF4 but only variation in the TaPIF4 promoter was shown. Is there any natural TaPIF4 alleles that have mutation on the phosphorylation sites?

7. The promoter mutated TaPIF4s were frequently found in Chinese varieties. The author interpreted this observation by involving the tightly linked gene, TaVRN-B1. This is a very interesting finding. The detailed sequence information of these two genes in the germplasm would help to support this hypothesis.

Minor:

1. The author should clarify the materials newly generated by this study and the materials inherited from their previous study (Chen et al., 2020).

2. Are the Tapif4 knockout lines T-DNA free?

3. Please repeat Figure 2D.

We have responded to all the comments raised by the reviewers as noted below. We would like to thank the editors and reviewers for their invaluable comments and suggestions that have helped to improve this manuscript. All the page number and line number mentioned below are based on **the manuscript of highlighted version**.

Reviewer #1 (Remarks to the Author):

The paper titled "Natural sequence variation of STKc_GSK3 kinase TaSG-D1 contributes to thermotolerance in Indian dwarf wheat" presents a study that identified that the mutation E286K on the gene *Tasg-D1* is responsible for the protein TaSG-D1's role in thermotolerance. The authors subsequently showed both WT (TaSG-D1) and the (amino acid substitution E286K), which the authors confusingly named *Tasg-D1*, suggesting this is a recessive allele - is it? unless I am missing something here as there is significant amount of genetic work in this study, including QTL).

Response: We apologize for the confusing nomenclature. It is a gain-of-function mutation and a semi-dominant allele, we used "*Tasg-D1*" to stand for the gene with E286K mutation contributing to enhanced thermotolerance, whereas "*TaSG-D1*" stands for the wild type. Here, we agree with the reviewer's suggestion in **Comment 1**, and redesigned the *Tasg-D1* gene as *TaSG-D1^{E286K}* in the revised manuscript.

The authors then demonstrated that both TaSG-D1 and *Tasg-D1* (here-forth E286K mutant to minimise confusion in this report) interacts with C-terminal of PIF4 through Yeast-2-hybrid, Co-PI and LCI assay in *N. benthamiana*. Additionally, this study demonstrated PIF4's role in thermotolerance through CRISPR-KO of wheat followed by transcriptomic studies on PIF4 mutants at higher temperatures - It is well known that that PIF4 is an important factor in plant thermotolerance. Although most work is limited to arabidopsis and it is nice see this confirmed in a monocot. The authors next demonstrated that both *tasgD1* and *taSG-D1* were found to trigger PIF4 phosphorylation. However, it is difficult to assess whether this is really case as I cannot see any obvious differences (Figure 3B and C).

Response: According to the reviewer's suggestions, we re-performed the phosphorylation experiments, and substituted the original **Figure 3B and 3C** with new ones in the revised manuscript (see **Response 2 below**).

We next move to the most interesting revelation where the authors revealed that mutation E286K is responsible for SG-D1 protein stability during heat stress at 42°C. The study ends with PIF4 loci analysis and the relationship of its mutations and geographical distribution. Overall, I think the main finding of this work important, that TaSG-D1 interacts with PIF4 and that mutation E286K contributes to the TaSG-D1 protein stability. However, the results at its current state is preliminary and is more appropriate for a specialised journal rather than Nature communications. Below are my major and minor comments

Major comments.

Comment 1. Naming E286K mutant of SG-D1 to *sg-D1* makes the whole manuscript extremely confusing to read. The authors should at least consider SG-D1-E286K.

Response 1. We agree with the reviewer's suggestion and redesigned the *Tasg-D1* as *TaSG-D1^{E286K}* in the revised manuscript.

Comment 2. It is nice that the author demonstrates both WT and E286K mutant interacts with PIF4 and that the authors tried to However, I find the claim that TaSG-D1 mediates phosphorylation of TaPIF4 unconvincing. Positive and negative controls are essential for this study (e.g. Figure 3B and C) - essentially all bands look the same to me whether in presence of ATP, either TaSG-D1, etc...

Response 2. According to the reviewer's suggestions, we re-performed the related phosphorylation experiments with modified parameters. The new **Fig. 3b** and **3c** show phosphorylated form and non-phosphorylated form of TaPIF4 protein simultaneously. In addition, we added Calf Intestinal Alkaline Phosphatase (CIAP) to prevent phosphorylation, which was often used as a negative control in phosphorylation assay. Moreover, we used positive controls (alpha-Casein) to validate that phos-tag works well to differentiate the phosphorylated and non-phosphorylated protein forms in our analysis (**Fig. 3b, 3c and Supplementary Fig. 6a**).

Comment 3. While I agree the mutation E286K do contribute to thermo-stability of TaSG-D1 protein, the results suggested the lower level of PIF4 associated (whether wt or mutant) simply correlates with protein abundance (i.e. less SG = less associated PIF4). It would be interesting for the authors to assess the total amount of PIF4 in comparison to SG-associated PIF4.

Response 3. We appreciate the reviewer's suggestions and analyzed the SG-associated TaPIF4 in comparison to the total amount of TaPIF4 by co-transforming TaPIF4 with TaSG-D1 and TaSG-D1^{E286K}, respectively. We found that 12% of TaPIF4 is detectable after 3-h heat stress in the control, whereas the proportions are 23% and 88% when co-transforming with TaSG-D1 and Ta-SG-D1^{E286K}, respectively, which indicate that 11% and 76% of TaPIF4 were associated with and protected by TaSG-D1 and TaSG-D1^{E286K}, respectively. We added the description and related figure (**Supplementary Fig. 9**) on page 6, line 169-176 in the revised manuscript as follows "Furthermore, we analyzed the TaSG-D1^{E286K}/TaSG-D1-associated TaPIF4 in comparison to the total amount of TaPIF4 in response to heat stress, and found that 23% and 88% of TaPIF4 protein were detectable after 3 h heat stress when co-transforming with TaSG-D1 and TaSG-D1^{E286K}, respectively, in *Nicotiana benthamiana* leaves, whereas only 12% of TaPIF4 protein remains stable in the control (**Supplementary Fig. 9**). These results suggest that 11% and 76% of TaPIF4 are associated with and protected by TaSG-D1 and TaSG-D1^{E286K}, respectively, after 3h heat treatment".

Comment 4. Kinase activity for *tasgD1* and *taSG-D1* -on PIF4 phosphorylation where are the controls? everything look the same to me.

- Can use a recombinant protein, phosphatase treated as control.
- some sort of quantification to transform the band intensity to bar plots (e.g. phos-tag vs anti-GST).

Response 4. According to the reviewer's suggestions, we re-performed the phosphorylation analysis with modified parameters, and added Calf Intestinal

Alkaline Phosphatase (CIAP) to prevent phosphorylation (new **Fig. 3b, 3c**), which was often used as negative control. Moreover, we used positive controls (alpha-Casein) to validate that phos-tag works well to differentiate the phosphorylated and non-phosphorylated protein forms in our analysis (**Supplementary Fig. 6a**).

In addition, we transformed the band intensity to bar plot to quantify phosphorylation intensity of TaPIF4 (**Supplementary Fig. 6b, 6c**).

Comment 5. Major rewrite of this manuscript is crucial. Currently it is rather eclectic, especially at most of the work is really on TaSG-D1 and its effect on PIF4, and therefore thermotolerance e.g. discuss TaSG-D1 distribution alongside PIF4 instead of PIF4 alone - while I appreciate the amount of work required to generate a wheat pif4 mutant, this paper is not about PIF4 alone.

Response 5. We fully understand the reviewer's concern that limited discussion on *TaSG-D1* is available. Thus, we studied the distribution of E286K variation of TaSG-D1 in global wheat genetic resources, and found that the E286K substitution is a rare variation, which is restricted to India and Pakistan accessions. This specific distribution is likely due to the high-temperature conditions in the Indian subcontinent during wheat growth period, where the introduction of E286K variation would be beneficial in overcoming this challenge. However, despite the improved thermotolerance, the E286K substitution in TaSG-D1 confers a negative impact on grain yield and has not been widely adopted after the Green Revolution. In addition, we also investigated the variations in the 2-kb promoter region of *TaSG-D1* during wheat breeding history, as determined by analyzing published resequencing data of 159 modern cultivars and 172 landrace accessions, and revealed that the promoter sequence is conserved and no genomic differentiation is observed between each other (**Supplementary Fig. 10**). We added the discussion on *TaSG-D1* on page 7, line 215-229 as follows "To better understand the utilization of the *TaSG-D1*^{E286K} allele during wheat breeding programs, we conducted an analysis of haplotype distribution among global wheat genetic resources according to published resequencing data¹⁷⁻²⁰ (**Supplementary Table 5**). Our findings indicate that the E286K variation is limited to *T. sphaerococcum* in India and Pakistan (**Supplementary Fig. 10**). This specific distribution is likely due to the high-temperature conditions in the Indian subcontinent during wheat growth period, where the introduction of E286K variation would be beneficial in overcoming this challenge. However, despite the improved thermotolerance, the E286K substitution in TaSG-D1 confers a negative impact on grain yield and has not been widely adopted after the Green Revolution. In addition, we also investigated the variations in the 2-kb promoter region of *TaSG-D1* during wheat breeding history, as determined by analyzing published resequencing data of 159 modern cultivars and 172 landrace accessions¹⁷⁻²⁰, and revealed that the promoter sequence is conserved and no genomic differentiation is observed between each other (**Supplementary Fig. 10**)".

Minor comment

Comment 6. please don't put band quantification on top of bands. Either include the values under the bands or plot it next to the gel images

Response 6. We revised the related figures and put the values under the bands in the

Reviewer #2 (Remarks to the Author):

Cao et al. identify in this work a Tasg-D1 thermotolerance gene in the Indian dwarf wheat (*Triticum sphaerococcum*), encoding a STKc_GSK3 kinase with an E286K substitution as compared to the TaSG-D1 wild type. This amino acid substitution stabilizes in heat stress conditions the Tasg-D1 protein and confers increased binding affinity towards its TaPIF4 target, causing elevated TaPIF4 phosphorylation levels and greater stability of the protein, which correlates with an enhanced thermotolerance of dwarf wheat. In line with this regulation, loss of function of Tasg-D1 or TaPIF4 impair wheat thermotolerance. Comparative genomic studies identified the presence of several InDels in the TaPIF4 promoter of modern Chinese wheat cultivars, leading to decreased TaPIF4 expression levels in response to heat stress. Variation in this region appears to have been introduced from European germplasm, while it is absent in local landraces. Those were likely co-retained during selection of the TaVrn-B1 superior allele, these findings hence uncovering TaPIF4 as a potential locus for wheat thermotolerance improvement.

Via QTL mapping of a *T. sphaerococcum* and *T. aestivum* RIL population, authors had earlier identified a *sphaerococcum* allele which carries a E286K substitution in the TaSG-D1 gene, a GSK3 kinase homologous to *Arabidopsis* BIN2, as conferring the semispherical grain morphology of Indian dwarf wheat. In this work authors show that this substitution is also responsible for enhanced thermotolerance of Indian dwarf wheat. NILTasg-D1 lines with this allele display a higher survival to 3 days 42°C treatments than near isogenic NILTaSG-D1 lines. Authors show that heat tolerance is suppressed in *sphaerococcum* M2-1 mutants, with a loss-of-function mutation in the Tasg-D1 gene, while a better heat stress growth performance is observed in Tasg-D1 OE lines. These findings strongly support the E286K substitution to account for the improved thermotolerance trait. A Y2H screen for Tasg-D1 interactors identified the wheat TaPIF4 factor, which is confirmed by Luciferase complementation, pull-down and co-IP assays. CRISPR-Cas9 knock-out mutations of TaPIF4 in the Fielder cultivar are here shown to lead to impaired thermotolerance, and to correlate with down-regulated expression of several “heat responsive”, “protein folding” and “ROS responsive” transcripts under heat stress conditions. Both Tasg-D1 and TaSG1-D1 were found to phosphorylate TaPIF4, allowing LC-MS/MS identification of 14 potential phosphorylation sites. Still, mutation of these residues reduces TaPIF4A phosphorylation, but did not impair in vitro phosphorylation of this protein by Tasg-D1. Phosphomimic Ser/Thr to Asp substitutions within these residues are however shown to enhance TaPIF4D heat stability, while this to be reduced in the TaPIF4A protein. Moreover, TaSG-D1 was observed to be progressively degraded under heat-stress conditions, while Tasg-D1 abundance remain unaffected. TaPIF4 is as well shown in cell-free degradation assays to be more stable in the presence of Tasg-D1 than TaSG-D1, suggesting Tasg-D1 stabilization contributes to higher TaPIF4 phosphorylation and to the accumulation of the protein under heat stress conditions, which correlates with improved thermotolerance.

The Tasg-D1 mutation negatively affects grain yield and was not introduced in Green Revolution cultivars. Remarkably, comparative genomics of modern and local China landraces identified a 25-Mb hypervariable footprint region around the TaPIF4 locus. Various InDels interrupting putative stress-response were indeed observed in the 2-kb TaPIF4 promoter region, which are absent in local landraces. Expression studies showed these changes reduce heat activation and thus may play a role in lowering thermotolerance of modern cultivars. The TaVrn-B1 gene locates in the 25-Mb footprint region, indicating that this region was co-introduced together with the TaVrn-B1 superior allele during modern wheat breeding. Modification of the TaPIF4 promoter is thus proposed as a target for improving wheat thermotolerance.

Overall, this is an exciting piece of work that provides solid evidence for the E286K substitution in the STKc_GSK3 T. sphaerococcum allele to confer enhanced tolerance to heat stress. Notably, this substitution locates into the same TREE domain as the Arabidopsis bin2.1 mutation described in Chory's lab (Li et al., 20021) and the ucu1 mutant identified in Micol's laboratory (Perez-Perez et al., 2002), and shown to encode a hypermorphic allele. Mutations in the TREE domain prevent BR-mediated BIN2 inhibition (Peng and Li, 2003) and were established to result in increased BIN2 stability (Vert and Chory 2006; Peng et al., 2008) as it is here shown for the wheat Tasg-D1 protein. Arabidopsis BIN2 is on the other side reported to phosphorylate and destabilize the Arabidopsis PIF4 and PIF3 transcription factors (Bernardo-García et al., 2014; Ling et al., 2017). Therefore, in opposite to the results reported here for wheat, phosphorylation was found in Arabidopsis to lead to the destabilization of both proteins. Such contrasting results might be indicative of a different behavior of the wheat PIF factors, or most likely that stability of these proteins is reversed under heat stress (42°C).

Response: According to the reviewer's suggestion, we analyzed the stability of TaPIF4 under both normal and heat stressed conditions, and found that Tasg-D1 (re-designed as TaSG-D1^{E286K} in the revised manuscript) confers enhanced TaPIF4 stability under both conditions (**Fig. 4b and Supplementary Fig. 8d**).

Indeed, ND4332 and NILTasg-D1 lines show a clear dwarf phenotype suggestive of a reduced stability of TaPIF4 under control conditions. Surprisingly, such dwarfing phenotype is less evident in Tasg-D1 OE lines. Additionally, a semidwarf phenotype is observed in Tapif4-2 knock-out mutants and to a lesser extent in the Tapif4-7 lines, again supporting a role of TaPIF4 in promoting cell elongation in a similar manner as shown in Arabidopsis (why do these lines display a different phenotype when both correspond to ko mutants?).

Response: Knockout of *TaPIF4* results in reduced plant height, indicating that *TaPIF4* is a positive regulator of wheat growth. However, our data demonstrated that Tasg-D1 (re-designed as TaSG-D1^{E286K} in the revised manuscript) confers enhanced TaPIF4 stability under both normal and heat stressed conditions, but leads to reduced plant height. This confusing result suggest that there might be other regulators involving in the regulation of plant height in wheat. Consistent with our hypothesis, a recent study demonstrated that Tasg-D1 (TaSG-D1^{E286K}) interacts with and

phosphorylates Rht-B1b, the Green Revolution gene in wheat, to reduce plant height in wheat (Dong et al., 2023), this signaling pathway might play a major role in regulating semi-dwarf phenotype in wheat.

Tasg-D1 (TaSG-D1^{E286K}) OE lines indeed exhibits dwarfing phenotype under normal conditions, we re-performed the analysis and confirmed the observation, and replaced the corresponding panel in the new **Fig. 1a (4th panel)**.

In the original **Fig. 3a**, *Tapif4-7* lines exhibit lesser extent of semidwarf phenotype compared with *Tapif4-2* probably due to inappropriate light conditions, because the leaf sheaths of these seedlings are curved. Thus, we re-performed the analysis, and found that *Tapif4-7* knockout lines show similar phenotypic variation in terms of plant height and thermotolerance. We replaced the corresponding figure (**Fig. 3a**) in the new version of manuscript.

As such, it would be critical to carry out similar cell-free degradation assays as shown in Figure 4, under control temperatures, and thus establish that BIN2-dependent control of PIFs stability is opposite under heat stress and control conditions, or rather than the wheat and Arabidopsis factors display an opposite response.

Response: We agree with the reviewer's suggestions, and performed the cell-free degradation assay to detect PIF4 stability under both normal and heat stressed conditions (**Fig. 4b and Supplementary Fig. 8d**). The results show that *Tasg-D1* (TaSG-D1^{E286K}) improves the stability of PIF4 under both conditions, which display an opposite response compared to Arabidopsis. This discrepancy may be due to the different phosphorylation sites of PIF4 in the two species, because our study identified 14 phosphorylation sites of TaPIF4 through LC-MS/MS in wheat, which differs from the previously reported three phosphorylation sites of PIF4 in Arabidopsis (*Arabidopsis thaliana*). Consistent with our hypothesis, it is reported that MPK3 can phosphorylate the Ser94 site of ICE1, leading to ICE1 degradation, and negatively regulating cold tolerance in Arabidopsis; However, OsMPK3 can phosphorylate the Thr404, Thr406, Ser407, Thr412, and Ser433 sites of OsICE1 and enhance its protein stability, thereby positively regulating cold tolerance in rice (*Oryza sativa*). We added the description in the revised manuscript **on page 6, line 179-189** as follows in the revised manuscript **"It is worthy noticing that BIN2, the homolog of TaSG-D1 in Arabidopsis (*Arabidopsis thaliana*), phosphorylates and destabilizes PIF4. This discrepancy may be due to the different phosphorylation sites of PIF4 in the two species, because our study identified 14 phosphorylation sites of TaPIF4 through LC-MS/MS in wheat, which differs from the previously identified three phosphorylation sites of PIF4 in Arabidopsis⁸. Consistent with our hypothesis, it is reported that MPK3 can phosphorylate the Ser94 site of ICE1, leading to ICE1 degradation, and negatively regulating cold tolerance in Arabidopsis⁹⁻¹¹; However, OsMPK3 can phosphorylate the Thr404, Thr406, Ser407, Thr412, and Ser433 sites of OsICE1 and enhance its protein stability, thereby positively regulating cold tolerance in rice (*Oryza sativa*)^{12,13"}.**

Although the findings nicely imply a role of TaPIF4, downstream of *Tasg-D1*, in

conferring thermotolerance, they do not exclude the participation of other transcriptional factors in conferring this trait. AtPIF4 was shown to heterodimerize with BES1/BZR1, and this complex to play a role in regulating BR-synthesis (Martínez et al., 2018). Authors should discuss this possibility and more importantly discuss their findings in the context of the previous knowledge generated in Arabidopsis.

Response: According to the reviewer's suggestions, we added more discussions in the revised manuscript on **page 6-7, line 198-212** as follows “**Although our study revealed that TaPIF4, as a downstream target of TaSG-D1^{E286K}, confers thermotolerance by regulating a set of heat response genes including HSPs, HSFs, and TaMBF1c in wheat, which is consistent with the finding in Arabidopsis that PIF4 induces the expression of HSF A2 by directly binding to its promoter, thereby triggering basal thermotolerance¹⁴, we should notice that other transcriptional factors may also be involved in regulating thermotolerance through the TaSG-D1 signaling pathway. Because PIF4 has been shown to interact with and modify the DNA binding specificity of BRI1-EMS-SUPPRESSOR 1 in Arabidopsis, leading to the switch of repressive activity to co-activation function, which in turn de-represses BR biosynthesis-related gene expression in response to warmth¹⁵. Moreover, BRASSINAZOLE RESISTANT 1, another transcription factor in Arabidopsis, accumulates in the nucleus under high temperature conditions and binds to the promoter of PIF4, inducing its expression¹⁶. Hence, it is possible that other factors, besides TaPIF4, play a role in the regulation of thermotolerance in wheat, and further research is needed to fully comprehend the underlying mechanisms”.**

Reviewer #3 (Remarks to the Author):

The authors present a study looking at natural sequence variations in a protein kinase that has ramifications for thermotolerance in wheat. The authors present a number of lines of evidence to suggest a mechanism by which a mutation in the protein kinase results in weaker degradation of PIF4 leading to thermotolerance.

While the authors present a reasonable story, there are substantial issues that need to be rectified.

Major issues:

Comment 1. Many of the figures are of substandard quality and need major attention.

a. multiple bar graphs depicting data without the presentation of replicates as is standard in publication. The box plot of Figure 1 is a good example that should be replicated across the manuscript.

Response 1a. According to the reviewer's suggestions, we re-plotted the bar graphs of **Fig. 1b, Fig. 3a, Fig. 5d, Supplementary Fig. 2b, Supplementary Fig. 6b, Supplementary Fig. 6c, and Supplementary Fig. 8a.**

b. Figure 4 and S6 have blot panels that are cropped too tightly with densitometry values overlaid atop the blot. The numerical values should be placed where they do not obstruct the blot. Further, each of the presented blot windows need to be widened.

Response 1b. According to the reviewer's suggestions, we placed the numerical values under the blot, and replaced the presented blot pictures with widened ones in

the related revised figures (new **Fig. 2, Fig. 3, Fig. 4, Supplementary Fig. 7, Supplementary Fig. 8, Supplementary Fig. 9**).

c. Figure S6c should have all panels the same size with corresponding molecular markers to be able to properly assess the size shift claimed. Further, from the blot presented it is unclear if there is in fact a distinct size shift as there is a large smear, unlike what is presented in Figure 3.

Response 1c. We re-performed the phosphorylation analysis of **Fig. 3b, 3c** and **Supplementary Fig. 6c (Supplementary Fig. 8c** in the revised manuscript) with modified parameters according to the suggestions, which presented both phosphorylated protein forms and non-phosphorylated protein forms. In addition, we displayed panels with similar size with corresponding molecular makers.

d. Figure 3 should be corroborated by an anti-phospho blot as it is not immediately clear if there is a size shift or if the blot ran awkwardly. PhosTag gels have a tendency to run awkwardly, so a blot of these samples would confirm the result.

Response 1d. We tried anti-phospho analysis using commercial antibody (Anti-Phospho-(Ser/Thr), abcam, ab117253), however, it does not work well in our experiments. We then re-performed the phosphorylation analysis using PhosTag gel (**Fig. 3b, 3c** and **Supplementary Fig. 8c** in the revised manuscript) with modified parameters, which presented phosphorylated protein forms and non-phosphorylated protein forms.

e. Figure 2: It is unclear which band is PIF4 or if these are degradation products or even non-specific binding of the antibody.

Response 1e. We performed the pull-down assay again, and marked the specific bands for TaPIF4 (new **Fig. 2c**), there are some non-specific bands in the pull down assay.

f. Full length protein stained gels and corresponding western blots should be provided in a supplemental figure to allow the reader to assess the purity of the preparations used in the assays.

Response 1f. We added the full length protein stained gels and corresponding western blots in **Source data** as the reviewer suggested.

g. Figure s5 the LCMSMS data is unreadable. This data should be assembled into a supplemental table that aligns with field standards for this data type with the corresponding data common to PTM site identification included (e.g. site localization reliability value, score, amongst multiple other data types).

Response 1g. We removed the original **Supplementary Figure 5** and added a **Supplemental table 4** to illustrate the phosphorylation information

h. Figure s4. It is unclear what background was used to perform the GO enrichment analysis and what the statistical threshold was and the correction used was.

Response 1h. With apologies for the unclear description, GO enrichment analysis was conducted by using the ClusterProfiler (v.3.12.0), all the annotated genes (107,891) in wheat reference genome were set as the background for hypergeometric test, and

Benjamini-Hochberg method was used for multiple testing correction of P values. Finally, the top 10 significantly enriched GO terms of Cellular component, Molecular function and Biological process with an adjusted P value < 0.001 were displayed in the heatmap.

i. All figures containing western blots do not provide any indication of replication nor how much sample was loaded per lane.

Response 1i. We added the related information in figure legends and methods section, and provided replicated figures in **Source data**.

j. A threshold for differential gene expression of $FDR < 0.05$ and fold change of < 0.5 is too loose. The FDR p-value is fine, but minimally a FC of < 1.5 is required to minimally align with field standards.

Response 1j. We apologized for the inappropriate description, we used the criteria with fold-change ≥ 2 and $FDR < 0.05$, we revised the statement on **page 4, line 117-120** of revised manuscript as follows “**In total, 2,289, 4,981 and 7,278 genes were down-regulated in *Tapif4-KO* line compared with WT ($\log_2[\text{fold change}] \leq -1$ and $FDR < 0.05$) at 0 h, 3 h and 6 h, respectively (Supplementary Table 3)**”.

k. Pictures of replicates for the LUC assay should be shown in a supplemental figure.

Response 1k. We provided pictures of replicates in **Source data**.

Comment 2. Multiple methods are inadequately described.

a. For the GO enrichment what was the FDR correction used? See other GO enrichment commentary above in Major Concern 1.

Response 2a. GO enrichment analysis was conducted by using the ClusterProfiler (v.3.12.0), all the annotated genes (107,891) in wheat reference genome were set as the background for hypergeometric test, and Benjamini-Hochberg method was used for multiple testing correction of P values. Finally, the top 10 significantly enriched GO terms of Cellular component, Molecular function and Biological process with an adjusted P value < 0.001 were displayed in the heatmap. We added the information in materials and methods section on **page 21, line 627-633**.

b. The LCMSMS methods are completely deficient. So much so, that it is too much to fully comment on. The authors need to consult other leading manuscripts in the field describing these methods and carefully describe what they did here. The current approach of citing another paper is inappropriate for methods this complex. While the results are intriguing, the aforementioned deficiencies in presentation, methods and results require substantial attention.

Response 2b. We agree with the reviewer and revised the description in the new version of manuscript on **page 22, line 671-689** as follows: “**We performed liquid chromatography-tandem mass spectrometry (LC-MS/MS) assay to identify putative phosphorylation sites of TaPIF4 according to past reports³⁸. GST-tagged TaPIF4 (100 ug) protein were co-incubated with His-tagged TaSG-D1 (50 ug) proteins in a kinase reaction buffer (25 mM Tris-HCl [pH 7.5], 12 mM MgCl₂, 5 mM ATP, and 1 mM DTT) at 37°C for 1 h. The total reaction proteins in vitro phosphorylation assays were reduced by DTT, alkylated by IAM, digested overnight at 25°C by trypsin and then**

diluted by 0.1% [v/v] formic acid. After centrifuge at 12,000 g for 20 minutes, the collected supernatant was analyzed by nano-Acquity nano HPLC (Waters, Milford, MA, USA) coupled with a Thermo Q-Exactive high resolution mass spectrometer (Thermo Scientific, Waltham, MA, USA) for LC-MS/MS assays. Then, the MASCOT search engine (Matrix Science, Mascot 2.6.0) was used to analyze the raw MS/MS data. The UniProt wheat sequence database was used to search peptides. The search parameters were employed as follows: trypsin was set as a specific enzyme, a maximum missed cleavages were set to 2, fixed modification contain carbamidomethylation (Cys) and variable modification contain oxidation (Met) and phosphorylation (Ser, Thr and Tyr). Peptide and fragment tolerances were 10 ppm and 0.02 Da. Threshold value of positive peptides was MASCOT score > 35 corresponding to false positive rate < 5%. The positive peptides and identified phosphorylation sites were listed in **Supplementary Table 4**".

Reviewer #4 (Remarks to the Author):

Heat stress is one of the major threatens for agriculture, especially under the condition of global warming. Dissection of the genetic basis of thermotolerance helps to breed heat resistant crops. However, not much progress has been made. In the manuscript by Cao et al., the author presented the cloning of a thermotolerance QTL gene, TaSG-D1. An amino acid substitution (E286K) in TaSG-D1 resulted in increased stability of the protein and stronger interaction with its target, TaPIF4, which then regulates targets like TaMBF1c to enhance heat tolerance. Interestingly, they found that several weak alleles of TaPIF4 (variations in promoter), instead of the wild type allele are widely spread in Chinese modern wheat cultivars and they proposed that this phenomenon is caused by the linkage with TaVrn-B1. Breaking this linkage drug may bring new opportunity to enhance heat resistance in wheat. Aa far as I know, this is the first report about cloning of heat resistant gene by forward genetic approach. I have several concerns before I could recommend it for publication.

Major comments:

Comment 1. The authors provided several lines of evidences like phenotype of OE lines and EMS mutant to approve the candidacy of TaSG-D1. However, in the concept of forward cloning, they are still indirect evidences. Complementary test or functional validation of both alleles is necessary to demonstrate that the phenotype variation of the NIL is caused by the candidate gene. Meanwhile, more information of the interval (size, genes existed) should be provided.

Response 1. We fully understand the reviewer's concern that complementary test or functional validation of both alleles is necessary to demonstrate the candidacy of TaSG-D1. Firstly, we proved that gain-of-function of the A homeolog of *TaSG-D1* gene also contributes to the enhanced thermotolerance in wheat on **page 3, line 87-90**, which is comparable to functional validation of two alleles. Secondly, we analyzed the phenotypes of transgenic lines with different expression levels of *Tasg-D1* (re-designed as *TaSG-D1^{E286K}* in the revised manuscript) under heat stressed conditions, and found that the thermotolerance is positively associated with the expression level

of *Tasg-D1* (*TaSG-D1*^{E286K}, new **Supplementary Fig. 3a**). Thirdly, we compared the thermotolerance of *Tasg-D1* (*TaSG-D1*^{E286K}) OE line with *TaSG-D1* OE line, which exhibited similar expression levels, and found that *TaSG-D1*^{E286K} OE line exhibited enhanced thermotolerance compared with *TaSG-D1* OE line (new **Supplementary Fig. 3b**). We believe that all these lines of evidences collectively demonstrated that *TaSG-D1*^{E286K} is the candidate gene for thermotolerance improvement in wheat. We added the information on **page 3, line 82-90** in the revised manuscript as follows “**and its expression levels were positively correlated with the SSRH (Supplementary Fig. 3a). Moreover, we compared the thermotolerance between *TaSG-D1* OE line and *TaSG-D1*^{E286K} OE line, which exhibited similar expression levels, and found that *TaSG-D1*^{E286K} OE line exhibited enhanced thermotolerance compared with *TaSG-D1* OE line (Supplementary Fig. 3b). In addition, IIA236, an EMS mutant line in NC4 background with a gain-of-function of *TaSG-A1* (E286K substitution in TraesCS3A02G136500, a homeolog of *TaSG-D1* on chromosome 3A), exhibited improved thermotolerance compared to the wild type (75.0% vs. 12.5% of SSRH, **Fig. 1a, 1b**)”.**

In addition, according to the reviewer’s suggestions, we provided more information for the genetic interval (**Supplementary Table 1**). The genetic interval obtained from QTL mapping is ~38.6 Mb in length (Chr3D: 76377242- Chr3D:115033012), and contains 292 high-confidence genes, among which, 20 genes process variations in coding sequence including the *TaSG-D1*. We added the information in the revised manuscript on **page 2 line 62-65**.

Comment 2. One interesting finding is that *Tasg-D1* has a strong binding ability to TaPIF4 than *TaSG-D1*. However, to say that *Tasg-D1* interacts more strongly with TaPIF4, only one LCI assay is insufficient, especially when there is no reference to perform calibration (Figure S6 B). Would it be possible to perform a competing interaction assay? For instance, equal amount of differentially tagged *Tasg-D1* and *TaSG-D1* could be co-incubated with TaPIF4 in one tube for later pull-down.

Response 2. We appreciated the reviewer’s suggestion, and performed a competing pull-down assay. *Tasg-D1* (*TaSG-D1*^{E286K} in the revised manuscript) and *TaSG-D1* were fused with *pET-32a* and *pET-28a* vector, respectively, enabling clear differentiation of the two proteins based on their molecular mass in western blot analysis. To perform the analysis, we added equal amounts of His-*Tasg-D1* (His-*TaSG-D1*^{E286K}, 60 KDa) and His-*TaSG-D1* (48 KDa) proteins to the solution containing GST-TaPIF4. This approach allowed us to detect the protein levels of both *TaSG-D1* and *Tasg-D1* (*TaSG-D1*^{E286K}) bound to TaPIF4 in a single reaction. Consistent with the results of the LCI assay, our findings indicate that *Tasg-D1* (*TaSG-D1*^{E286K}) has a stronger binding affinity to TaPIF4 than *TaSG-D1* (new **Supplementary Fig. 8b**). We have included the new figure in the revised manuscript on **page 56, line 1239**.

Comment 3. *Tasg-D1* and TaPIF4 are working together to confer heat tolerance but both *Tapif4* (knock out line) and *TasgD1* (functional allele) showed a semi-dwarf phenotype. Any explanation for this?

Response 3. Knockout of *TaPIF4* results in reduced plant height, indicating that *TaPIF4* is a positive regulator of plant growth. However, *Tasg-D1* (*TaSG-D1*^{E286K}) confers enhanced *TaPIF4* stability but leads to reduced plant height, this confusing result suggests that other regulators may be involved in the regulation of plant height in wheat. Recent studies have supported this hypothesis by demonstrating that *Tasg-D1* (*TaSG-D1*^{E286K}) interacts with and phosphorylates *Rht-B1b*, the Green Revolution gene, to reduce plant height in wheat (Dong et al., 2023). This signaling pathway may play a major role in regulating semi-dwarf phenotype in wheat.

Comment 4. The time series RNA-seq could provide more information but the data analysis is a bit too weak. At least, the author could calculate how many DEGs contain PIF4 binding motif. Validation for binding of *TaPIF4* to *TaMBF1c* (by EMSA or other means) is required.

Response 4. According to the reviewer's suggestions, we re-analyzed the RNA-seq data and provided more information about DEGs. According to the fimo software analysis with P value < 0.001 , we identified 1299, 2809 and 3912 down-regulated genes with *TaPIF4* binding motif (E-box motif) at 0 h, 3 h and 6 h after heat stress, which accounted for 56.7%, 56.4% and 53.8%, respectively. GO enrichment analysis showed that the down-regulated genes with E-box were involved in photosynthesis related terms under normal conditions, whereas those ones at 3 h and 6 h were predominantly enriched in "response to heat" and "unfolded protein binding" etc. We added the information on **page 4, line 129-135** as "Moreover, we identified 1,299, 2,809 and 3,912 down-regulated genes with *TaPIF4* binding motif (E-box motif) in their 1-kb promoter sequence at 0 h, 3 h and 6 h after heat stress according to the fimo software analysis (P value < 0.001), which accounted for 56.7%, 56.4% and 53.8% of down-regulated genes, respectively, and GO enrichment analysis of these genes showed similar results to that of down-regulated genes (**Supplementary Fig. 5c and Supplementary Table 3**)".

Moreover, we performed EMSA assay and proved that *TaPIF4* can bind to the E-box in the promoter region of *TaMBF1c* (new **Supplementary Fig. 5b**). We added the corresponding statement on **page 4, line 127-129** of revised manuscript as "the binding ability was confirmed by Electrophoretic mobility shift assay (EMSA) assay (**Supplementary Fig. 5b**)".

Comment 5. Due to negative effect on seed size, *Tasg-D1* has not been widely used. I am wondering whether there are other types of variations than E286K on *TaSG-D1* gene existing in the germplasm, particularly in promoter? By the way, is expression level correlated with the performance of *Tasg-D1* (maybe more *Tasg-D1* OE lines) in response to heat stress?

Response 5. According to the reviewer's suggestions, we investigated the sequence variations in the 2-kb promoter region of *TaSG-D1* during wheat breeding history, as determined by analyzing published resequencing data of 159 modern cultivars and 172 landrace accessions. We revealed that the promoter sequence is highly conserved and no significant genomic differentiation is observed between modern cultivars and landrace accessions (**Supplementary Fig. 10**). We added more discussion on *TaSG-*

D1 on page 7, line 225-229 as follows: “In addition, we also investigated the variations in the 2-kb promoter region of *TaSG-D1* during wheat breeding history, as determined by analyzing published resequencing data of 159 modern cultivars and 172 landrace accessions¹⁷⁻²⁰, and revealed that the promoter sequence is conserved and no genomic differentiation is observed between each other (Supplementary Fig. 10)”.

In addition, we analyzed the phenotypes of transgenic lines with different expression levels of *Tasg-D1* (*TaSG-D1*^{E286K}) under heat stressed conditions, and found that the thermotolerance is positively associated with the expression levels of *Tasg-D1* (*TaSG-D1*^{E286K}, new Supplementary Fig. 3). We added the corresponding statement on page 3, line 82-86 of revised manuscript.

Comment 6. *Tasg-D1* phosphonates TaPIF4 but only variation in the TaPIF4 promoter was shown. Is there any natural TaPIF4 alleles that have mutation on the phosphorylation sites?

Response 6. We analyzed the sequence variation on the phosphorylation sites of TaPIF4 using 159 modern cultivars and 172 landrace accessions, but did not found natural variations in the phosphorylation sites.

Comment 7. The promoter mutated TaPIF4s were frequently found in Chinese varieties. The author interpreted this observation by involving the tightly linked gene, TaVRN-B1. This is a very interesting finding. The detailed sequence information of these two genes in the germplasm would help to support this hypothesis.

Response 7. According to the reviewer’s suggestions, we analyzed the sequence variations of *TaPIF4* and *TaVRN-B1*. We identified two types of *TaVRN-B1* variations including short deletion (~2700 bp) and long deletion (~7300 bp, including the ~2700 bp deletion) in the first intron. Interestingly, we found that 87.5% wheat germplasm containing deletions in *TaVRN-B1* possess sequence variations in *TaPIF4* promoter, which supports our hypothesis that the prevalence of *TaPIF4* promoter variations in Chinese cultivars is probably an incidental by-product during artificial selection of *TaVrn-B1* superior allele in modern wheat breeding process. However, we also noticed that a large portion of Chinese cultivars only contain sequence variations in *TaPIF4* but not in *TaVRN-B1*, this might be because breeders tend to explore *TaVRN-A1* variations to control flowering time during Chinese breeding history (Zhang et al., 2008; Yan et al., 2003; Zhang et al., 2010), whereas retention of sequence deletions in *TaPIF4* promoter indicates its involvement in regulating other agronomic traits, which merits further study. We added the information on page 9, line 276-282.

Minor:

Comment 8. The author should clarify the materials newly generated by this study and the materials inherited from their previous study (Chen et al., 2020).

Response 8. We added detailed information in the materials and method on page 17, line 498-502.

Comment 9. Are the *Tapif4* knockout lines T-DNA free?

Response 9. We analyzed the *Tapif4* mutants by using specific primer pairs and found

the knockout lines are not Cas9 free, see **Figure below**.

Comment 10. Please repeat Figure 2D.

Response 10. We re-performed the experiment and replaced original **Figure 2D**.

References in response:

1. Dong, H. et al. GSK3 phosphorylates and regulates the Green Revolution protein Rht-B1b to reduce plant height in wheat. *The Plant Cell* koad090 (2023) doi:10.1093/plcell/koad090.
2. Zhang, X. K. et al. Allelic Variation at the Vernalization Genes *Vrn-A1*, *Vrn-B1*, *Vrn-D1*, and *Vrn-B3* in Chinese Wheat Cultivars and Their Association with Growth Habit. *Crop Science* **48**, 458–470 (2008).
3. Yan, L. et al. Positional cloning of the wheat vernalization gene *VRN1*. *Proc. Natl. Acad. Sci. U.S.A.* **100**, 6263–6268 (2003).
4. Zhang, Y. et al. Distribution and Selective Effects of *Vrn-A1*, *Vrn-B1*, and *Vrn-D1* Genes in Derivative Varieties from Four Cornerstone Breeding Parents of Wheat in China. *Agricultural Sciences in China* **9**, 1389–1399 (2010).

Reviewers' Comments:

Reviewer #1:

Remarks to the Author:

The authors have made significant improvements to the manuscript, resulting in clear and concise results. My only remaining comment is a suggestion that could enhance the discussion. Currently, there is an imbalance in the discussion between TaPIF4 and TaSG-D1, and the manuscript could benefit from a more comprehensive discussion, particularly regarding the analysis of TaSG-D1. Personally, I find the revelation that the E286K substitution is limited to Indian and Pakistani accessions (under high-temperature conditions) intriguing, especially considering that this substitution negatively affects grain yield (indicating a fitness cost of thermotolerance). This observation deserves a more in-depth discussion in conjunction with the distribution of TaPIF4. I suggest including SF10 in the main Figure 5 and moving the discussion on TaSG-D1 (lines 215-229) towards the end of the discussion, within the context of the global distribution of PIF4.

Reviewer #2:

Remarks to the Author:

Cao et al report in this work a role of the TaSG-D1E286K substitution earlier identified in Indian dwarf wheat (*Triticum sphaerococcum*) in conferring heat stress tolerance. The TaSG-D1E286K protein is shown in this work to bind with higher affinity the TaPIF4 factor, enhanced TaPIF4 phosphorylation resulting in its stabilization and in promoting heat stress survival. Authors show Chinese wheat cultivars to include TaPIF4 alleles showing promoter polymorphisms that drive reduced TaPIF4 expression levels, hence correlating with a higher susceptibility of these cultivars to heat stress. In this revised form authors addressed most of the concerns raised by the reviewers and highly improved quality of the phosphor-tag and co-IP gels, in addition to include further supportive data showing that heat stress tolerance is indeed contributed by the TaSG-D1E286K locus. Altogether, the work nicely demonstrates that the TaSG-D1E286K allele, equivalent to the Arabidopsis bin2-1 mutant, has a central role in heat stress tolerance and it mediates this response in part by modulating TaPIF4 phosphorylation and protein stability. However, while the E286K mutation causes a strong stabilization of the TaSG-D1E286K protein, its stabilizing effects on TaPIF4 are less robust and variable between independent experiments. To my opinion, authors discovered a novel function of TaPIF4 in modulating response to heat stress which involves a different regulatory mechanism than the thermal elongation role of PIF4. Authors use the term thermotolerance for their observations, and this is somehow misleading as it brings to believe that it involves the described thermomorphogenic role of PIF4. 42°C induce a heat stress response, as indicated by the GO terms "protein folding" and "response to ROS", rather than auxin and cell-wall related genes. Therefore, it would be more clarifying using the term heat stress tolerance in the title. The work proves that TaPIF4 is involved in this heat stress protection, but stabilizing effects of TaSG-D1E286K on the TaPIF4 protein are relatively mild as compared to the stabilizing effects of this mutation on TaSG-D1 itself. A simple model to explain these observations is that PIF4 mediates this heat stress response in concert with a yet unknown factor which is also regulated by BIN2. TaPIF4 is needed to this response as demonstrated by the observation that TaPIF4 knock-out lines are more susceptible to heat stress. However, the work does not demonstrate that stabilization of TaPIF4 is sufficient for heat stress tolerance. For that, it would be required showing that TaPIF4-D over-expressers are heat stress tolerant. The participation of additional TaPIF4 partners is already suggested in the MS, but significance of the work will be strongly increased if it is further highlighted, in addition to stressing that this response is different from thermomorphogenesis, and presumably involves a yet unknown PIF4 partner.

Minor comments:

Supplementary Figure 3. Correct abscissa label to Relative expression. Figure shows that survival is enhanced by increased levels expression of TaSG-D1E286K, whereas some protection is also attained by over-expression of the WT TaSG-D1 protein, as expected. However, stabilization of TaPIF4 in TaSG-

D1E286K lines would be foreseen to lead to taller plants while from the pictures these are observed to be shorter. Although results are fully consistent with the TaSG-D1E286K allele conferring increased heat stress tolerance, I am not convinced this effect is only mediated by TaPIF4 or regulated by the same regulatory network involved in thermal elongation. Authors should better differentiate the role of PIFs in heat stress and elevated ambient temperatures. That is, PIFs promote thermal elongation at warm temperatures but protect from oxidative damage under heat stress, presumably by interacting with additional unknown partners into a regulatory cascade basically different from thermal elongation. Therefore, it is better to refer to this response as heat stress tolerance to avoid confusion with the known function of PIFs in thermoelongation.

Supplementary Figure 4: a. Correct to "Normal" conditions. Temperature treatments in this study are milder than the 42°C heat stress treatments used in the rest of the work. NIL-TaSG-D1E286K plants display in this Figure a dwarf phenotype that is absent in TaSG-D1 lines. This would indicate that under milder temperatures TaSG-D1E286K behaves as the bin2-1 Arabidopsis gain of function allele in suppressing growth. This is however not supported by the finding that TaSG-D1 E286K stabilizes PIF4 also at normal temperatures. Then, why are these NIL plants shorter than NILs with the wild-type allele?.

Supplementary Figure 5: GO relates with protein folding, ROS response and heat stress. Such transcriptional changes are different from those related with auxin signaling and cell wall remodeling downstream of PIFs, and thus it can be concluded that this is a different mechanism to the PIFs thermal elongation pathway.

Supplementary Figure 7: Changes in protein stability are a lot stronger for TaSG-D1/TaSG-D1E286K as compared to TaPIF4. TaPIF4 is actually stabilized in the presence of TaSG-D1E286K but this effect is rather mild. BIN2 is recently known to phosphorylate multiple targets (Kim et al., Plant Cell 2023) and additional partners may in fact participate in the protective response to heat stress. This is discussed as a possibility, but it needs to be included as part of the proposed model.

Supplementary Figure 8: TaSG-D1E286K phosphorylates more strongly TaPIF4 (c). TaPIF4 is stabilized in the presence of TaSG-D1E286K but increase in protein stability is less than 2-fold (d) suggesting this to be an indirect effect.

Supplementary Figure 9: Differences in protein stability are here bigger than in Figure 8. Such differences may be due to the longer treatment at 42°C (3 h versus 30 min), but a proper control using the TaPIF4D protein would here be helpful to corroborate this regulation.

Reviewer #3:

Remarks to the Author:

The authors have addressed all my concerns.

Reviewer #4:

Remarks to the Author:

The authors have done nice job to address my concerns. However, one more question and one suggestion for the authors:

Question: The author provided the information of the QTL interval, in which 292 genes are identified. What about the internal of the NILs (heterozygous region)? This is more promising.

Suggestion: Please involve the information from the rebuttal (e.g. the semi-dwarf phenotype of pif4 and TasgD1) in the revised version.

We have responded to all the comments raised by the reviewers as noted below. We would like to thank the editors and reviewers for their invaluable comments and suggestions that have helped to improve this manuscript. All the page number and line number mentioned below are based on **the manuscript of highlighted version**.

Reviewer #1 (Remarks to the Author):

Comment: The authors have made significant improvements to the manuscript, resulting in clear and concise results. My only remaining comment is a suggestion that could enhance the discussion. Personally, I find the revelation that the E286K substitution is limited to Indian and Pakistani accessions (under high-temperature conditions) intriguing, especially considering that this substitution negatively affects grain yield (indicating a fitness cost of thermotolerance). This observation deserves a more in-depth discussion in conjunction with the distribution of TaPIF4. I suggest including SF10 in the main Figure 5 and moving the discussion on TaSG-D1 (lines 215-229) towards the end of the discussion, within the context of the global distribution of PIF4.

Response: We greatly appreciate the reviewer's suggestions, and revised the manuscript accordingly. We moved original **Supplemental Fig. 10** to the main **Fig. 5a**. Moreover, to enhance the discussion, we have introduced a new discussion section dedicated to exploring several key aspects **in line 300-371, page 10-12**. First, we delve into the fitness cost associated with heat tolerance in conjunction with the distribution of different haplotypes of TaSG-D1 and TaPIF4. Secondly, we proposed that other TaSG-D1-dependent regulators are probably involved in the regulation of heat stress tolerance. Then, we discuss the distinct phosphorylation sites of TaPIF4 that are triggered by TaSG-D1/BIN2, highlighting the discrepancies between wheat and Arabidopsis. Lastly, we discuss the potential reasons that why increased stability of TaPIF4 protein in TaSG-D1^{E286K} lines does not lead to increased height in wheat.

Reviewer #2 (Remarks to the Author):

Cao et al report in this work a role of the TaSG-D1E286K substitution earlier identified in Indian dwarf wheat (*Triticum sphaerococcum*) in conferring heat stress tolerance. The TaSG-D1E286K protein is shown in this work to bind with higher affinity the TaPIF4 factor, enhanced TaPIF4 phosphorylation resulting in its stabilization and in promoting heat stress survival. Authors show Chinese wheat cultivars to include TaPIF4 alleles showing promoter polymorphisms that drive reduced TaPIF4 expression levels, hence correlating with a higher susceptibility of these cultivars to heat stress.

In this revised form authors addressed most of the concerns raised by the reviewers and highly improved quality of the phosphor-tag and co-IP gels, in addition to include further supportive data showing that heat stress tolerance is indeed contributed by the TaSG-D1E286K locus. Altogether, the work nicely demonstrates that the TaSG-D1E286K allele, equivalent to the Arabidopsis bin2-1 mutant, has a central role in heat stress tolerance and it mediates this response in part by modulating TaPIF4 phosphorylation and protein stability.

Comment 1. However, while the E286K mutation causes a strong stabilization of the TaSG-D1E286K protein, its stabilizing effects on TaPIF4 are less robust and variable between independent experiments.

Response 1. We fully understand the reviewer's concern. We agree that the E286K mutation strongly stabilizes the TaSG-D1 protein, but its stabilizing effects on TaPIF4 are comparatively weaker. TaSG-D1 is a functional protein known to target multiple proteins involved in developmental regulation and defense responses¹⁻⁴. Given its potential to interact with and phosphorylate numerous proteins, it is reasonable to expect a less pronounced effect on stabilizing TaPIF4 compared to its own stability.

Regarding the variable stabilization effects of TaSG-D1 on TaPIF4, as evident in the original **Supplementary Figure 8d** and **Supplementary Figure 9**, we speculate that this discrepancy may derive from differences in treatment time between in vivo and in vitro degradation experiments. To address this concern, we conducted new in vitro protein degradation experiments with extended treatment at 42°C (60 min). The results demonstrated that TaPIF4 protein exhibited significant degradation in the absence of TaSG-D1/TaSG-D1^{E286K} protein, with only approximately 7% of the protein remaining. In contrast, when TaSG-D1 and TaSG-D1^{E286K} was present, the percentage of retained TaPIF4 protein increased to 25% and 52%, respectively (new **Supplementary Figure 10d**). Furthermore, we included TaPIF4D (the phosphorylation-mimic form of TaPIF4) as a control, as suggested below. Notably, the results showed consistency with the observations in the presence of TaSG-D1^{E286K}, with approximately 54% of the protein being retained (new **Supplementary Figure 10d**). Overall, the protein degradation trends, both in vitro (new **Supplementary Figure 10d**) and in vivo (new **Supplementary Figure 11**) experiments, exhibit comparability, albeit with some variations.

Comment 2. To my opinion, authors discovered a novel function of TaPIF4 in modulating response to heat stress which involves a different regulatory mechanism than the thermal elongation role of PIF4. Authors use the term thermotolerance for their observations, and this is somehow misleading as it brings to believe that it involves the described thermomorphogenic role of PIF4. 42°C induce a heat stress response, as indicated by the GO terms “protein folding” and “response to ROS”, rather than auxin and cell-wall related genes. Therefore, it would be more clarifying using the term heat stress tolerance in the title.

Response 2. We appreciate the reviewer's suggestion and revised the manuscript title as "Natural sequence variation of STKc_GSK3 kinase TaSG-D1 contributes to heat stress tolerance in Indian dwarf wheat". Additionally, we have made corresponding revisions throughout the whole manuscript to align with this updated terminology.

Comment 3. The work proves that TaPIF4 is involved in this heat stress protection, but stabilizing effects of TaSG-D1E286K on the TaPIF4 protein are relatively mild as compared to the stabilizing effects of this mutation on TaSG-D1 itself. A simple model to explain these observations is that PIF4 mediates this heat stress response in concert with a yet unknown factor which is also regulated by BIN2.

Response 3. Yes. Based on our calculations of seedling survival rates under heat

stress conditions, we observed that near-isogenic lines (NILs) of NIL^{TaSG-D1E286K} exhibited significantly enhanced heat tolerance compared to NIL^{TaSG-D1}, with survival rates of 73.6% and 4.2%, respectively. Whereas the survival rates for the wild type and *Tapif4* knockout mutants were 72.9% and 26.0%, respectively. These findings strongly suggest the involvement of unknown factors that act as downstream targets of *TaSG-D1*^{E286K} in the wheat heat stress response.

To address this important observation, we have included a discussion in the manuscript **in line 327-341, page 10-11**, emphasizing the potential presence of unidentified elements participating in the heat stress response in wheat. Furthermore, we have incorporated this information into the working model of the TaSG-D1/TaSG-D1^{E286K} signaling pathway in **Figure 4c** to illustrate their role in the response to heat stress.

Comment 4. TaPIF4 is needed to this response as demonstrated by the observation that TaPIF4 knock-out lines are more susceptible to heat stress. However, the work does not demonstrate that stabilization of TaPIF4 is sufficient for heat stress tolerance. For that, it would be required showing that TaPIF4-D over-expressers are heat stress tolerant.

Response 4. We deeply appreciate the reviewer's concern regarding the sufficiency of *TaPIF4* for heat stress tolerance in wheat, particularly using TaPIF4-D over-expressers. To address this concern, **(1)** we have made a request to Kong's lab⁵ for seeds of *TaPIL1/TaPIF4* overexpression lines, and examined their heat stress tolerance. As expected, *TaPIF4* overexpression lines exhibited improved heat tolerance compared to WT (31.25% vs. 68.75% vs. 70.83% for WT [KN199] vs. *TaPIF4-OE #1* vs. *TaPIF4-OE #2*, **Supplementary Fig. 5**). We added the corresponding description **in line 116-118, page 4**. **(2)** To validate the biological significance of phosphorylation-mimic form of *TaPIF4*, we overexpressed *TaPIF4D* in wheat, and found that *TaPIF4D* overexpressors also exhibited higher seedling survival rate than that of controls under heat stress conditions (29.17 % vs. 79.17 % vs. 81.25 % for WT [Fielder] vs. *TaPIF4D-OE-3* vs. *TaPIF4D-OE-9*, **Supplementary Fig. 9b**). Moreover, we observed that phosphorylation-mimic form of *TaPIF4* (*TaPIF4D*) overexpression lines exhibited enhanced heat stress tolerance compared with *TaPIF4* overexpression lines, which showed comparable expression levels of *TaPIF4D* and *TaPIF4*, in terms of differential seedling survival rate changes compared with their respective controls (37.50 % and 39.58 % increase for *TaPIF4* overexpression lines vs. 50.0 % and 52.08 % increase for *TaPIF4D* overexpression lines, **Supplementary Fig. 5 and Supplementary Fig. 9b**). These findings suggest that TaPIF4D is sufficient to enhance heat stress tolerance in wheat. We added the corresponding description **in line 159-170, page 5-6**.

Comment 5. The participation of additional TaPIF4 partners is already suggested in the MS, but significance of the work will be strongly increased if it is further highlighted, in addition to stressing that this response is different from thermomorphogenesis, and presumably involves a yet unknown PIF4 partner.

Response: According to the reviewer's suggestions, we have made several revisions

to our manuscript in **line 320-341, page 10-11**. Firstly, we have incorporated a discussion section on the role of *PIF4* in thermomorphogenesis in Arabidopsis. Additionally, we highlighted the distinctions between thermomorphogenesis and heat tolerance in this study. Moreover, we mentioned that unidentified partners of PIF4 might be involved in response to heat stress. Furthermore, we have updated our model diagram (**Fig. 4c**) to include other potential factors involved in heat stress tolerance in addition to TaPIF4.

Minor comments:

Comment 6. Supplementary Figure 3. Correct abscissa label to Relative expression.

Response 6. We have corrected abscissa label to “Relative expression” in the new **Supplementary Figure 3**.

Comment 7. Figure shows that survival is enhanced by increased levels expression of TaSG- D1E286K, whereas some protection is also attained by over-expression of the WT TaSG-D1 protein, as expected. However, stabilization of TaPIF4 in TaSG-D1E286K lines would be foreseen to lead to taller plants while from the pictures these are observed to be shorter. Although results are fully consistent with the TaSG-D1E286K allele conferring increased heat stress tolerance, I am not convinced this effect is only mediated by TaPIF4 or regulated by the same regulatory network involved in thermal elongation. Authors should better differentiate the role of PIFs in heat stress and elevated ambient temperatures. That is, PIFs promote thermal elongation at warm temperatures but protect from oxidative damage under heat stress, presumably by interacting with additional unknown partners into a regulatory cascade basically different from thermal elongation. Therefore, it is better to refer to this response as heat stress tolerance to avoid confusion with the known function of PIFs in thermoelongation.

Response 7. Thank you for the valuable suggestions. Previous studies have demonstrated that increased accumulation of PIF4 contributes to enhanced hypocotyl elongation in Arabidopsis by activating genes involved in the auxin pathway and cell wall organization⁶⁻¹¹. In our study, we revealed that knockout of TaPIF4 in wheat leads to a semi-dwarf phenotype, while overexpression of TaPIF4 or TaPIF4D increases plant height, consistent with the observation in previous study⁵.

In this study, however, the E286K substitution in TaSG-D1 increases the stability of TaPIF4 through phosphorylation, yet it paradoxically reduces plant height in wheat. Additionally, a previous study showed that overexpression of the non-phosphorylated form of the PIF4 protein (which exhibits enhanced protein stability) cannot restore the reduced plant height caused by the gain-of-function mutation of BIN2 (bin2-1, similar to TaSG-D1^{E286K})¹². These findings suggest the involvement of additional regulators in the regulation of plant height via the BIN2 pathway in plants.

Consistent with our hypothesis, a recent study demonstrated that TaSG-D1^{E286K} phosphorylates and stabilizes the DELLA protein Rht-B1b, which is known as the Green Revolution gene in wheat. Subsequently, the Rht-B1b protein interacts with TaPIF4, inhibiting its transcriptional activity towards downstream targets during plant

growth and ultimately reducing plant height in wheat⁵. Moreover, it has been reported that DELLA proteins interact with BZR1 and suppress its transcriptional activity, leading to growth inhibition in Arabidopsis¹³. Therefore, even though the stability of TaPIF4 protein is enhanced in TaSG-D1^{E286K} lines, its activity might be inhibited during plant growth, resulting in decreased plant height. This provides insights into the intricate molecular mechanisms underlying plant growth and development. We added the discussion in the revised manuscript (**line 354-367, page 11-12**).

In addition, we strongly agree with the reviewer's suggestion that we discovered a novel function of TaPIF4 in modulating response to heat stress which involves a different regulatory mechanism than the thermal elongation role of PIF4. We refer to this response as heat stress tolerance in the revised manuscript.

Comment 8. Supplementary Figure 4: a. Correct to "Normal" conditions.

Response 8. We corrected to "Normal conditions" in the new **Supplementary Figure 4**.

Comment 9. Temperature treatments in this study are milder than the 42°C heat stress treatments used in the rest of the work. NIL-TaSG-D1E286K plants display in this Figure a dwarf phenotype that is absent in TaSG-D1 lines. This would indicate that under milder temperatures TaSG-D1E286K behaves as the bin2-1 Arabidopsis gain of function allele in suppressing growth. This is however not supported by the finding that TaSG-D1 E286K stabilizes PIF4 also at normal temperatures. Then, why are these NIL plants shorter than NILs with the wild-type allele?

Response 9. Although TaSG-D1^{E286K} can phosphorylate and stabilize TaPIF4 under both normal and heat treatment conditions, but TaSG-D1^{E286K} lines paradoxically exhibit reduced plant height in wheat. These findings indicate that other genes are involved in the regulation of plant height in the TaSG-D1^{E286K} and TaPIF4 signaling pathway. Consistently, TaSG-D1^{E286K} phosphorylates and stabilizes the DELLA protein Rht-B1b, which is known as the Green Revolution gene in wheat. Subsequently, the Rht-B1b protein interacts with TaPIF4, inhibiting its transcriptional activity towards downstream targets and ultimately reducing plant height in wheat⁵. Moreover, it has been reported that DELLA proteins interact with BZR1 and suppress its transcriptional activity, leading to growth inhibition in Arabidopsis¹³. Therefore, even though the stability of TaPIF4 protein is enhanced in NIL lines with TaSG-D1^{E286K} lines, its transcriptional activity is inhibited, resulting in decreased plant height of NIL-TaSG-D1^{E286K} than that NIL-TaSG-D1 lines. We added the discussion in the revised manuscript (**line 354-367, page 11-12**).

These findings indicate that in addition to the regulation of TaPIF4 stability, the interaction between TaSG-D1^{E286K} and the DELLA protein Rht-B1b plays a crucial role in determining plant height in wheat. This highlights the complex interplay of multiple genes and proteins in the regulation of plant growth and development.

Comment 10. Supplementary Figure 5: GO relates with protein folding, ROS response and heat stress. Such transcriptional changes are different from those related with auxin signaling and cell wall remodeling downstream of PIFs, and thus it can be concluded that this is a different mechanism to the PIFs thermal elongation pathway.

Response 10. We agree with the reviewer's suggestions that our findings reveal a novel mechanism of *TaPIF4* in regulating the heat stress response, distinct from the well-known thermal elongation pathway. Considering this, we have appropriately modified the terminology in the revised manuscript to refer to this response as "heat stress tolerance". This revised terminology accurately reflects the unique regulatory mechanism by which *TaPIF4* contributes to the plant's ability to withstand heat stress. Thank you for bringing this to our attention, and we have made the necessary adjustments in the manuscript accordingly, and added related statements in the discussion section in **line 320-327, page 10**.

Comment 11. Supplementary Figure 7: Changes in protein stability are a lot stronger for *TaSG-D1/TaSG-D1E286K* as compared to *TaPIF4*. *TaPIF4* is actually stabilized in the presence of *TaSG-D1E286K* but this effect is rather mild. *BIN2* is recently known to phosphorylate multiple targets (Kim et al., Plant Cell 2023) and additional partners may in fact participate in the protective response to heat stress. This is discussed as a possibility, but it needs to be included as part of the proposed model.

Response 11. *TaSG-D1* is a functional protein known to target multiple proteins involved in developmental regulation and defense responses¹⁻⁴. Given its potential to phosphorylate and interact with numerous proteins, it is reasonable to expect a less pronounced effect on stabilizing *TaPIF4* compared to its own stability. According to the reviewer's suggestion, we added more discussion in **line 327-341, page 10-11** and revised the proposed model in **Fig. 4c** in the new version of manuscript.

Comment 12. Supplementary Figure 8: *TaSG-D1E286K* phosphorylates more strongly *TaPIF4* (c). *TaPIF4* is stabilized in the presence of *TaSG-D1E286K* but increase in protein stability is less than 2-fold (d) suggesting this to be an indirect effect.

Response 12. We understand the reviewer's concern. We conducted new in vitro protein degradation experiments with extended treatment at 42°C (60 min). The results indicate that without the presence of *TaSG-D1^{E286K}* protein, *TaPIF4* experiences significant degradation (~7% retained). In contrast, both *TaSG-D1* and *TaSG-D1^{E286K}* noticeably suppress degradation, resulting in 25% and 52% retention of *TaPIF4* protein, respectively (new **Supplementary Figure 10d**). Furthermore, we included *TaPIF4D* (phosphorylation form) as a control, which consistently yielded similar results to the inclusion of *TaSG-D1^{E286K}* (54% protein retained) (new **Supplementary Figure 10d**). But we agree with the reviewer's comment that it could be an indirect effect, and other mechanism, e.g. ubiquitylation, might be involved in the regulation of *TaPIF4* degradation, which merits further study.

Comment 13. Supplementary Figure 9: Differences in protein stability are here bigger than in Figure 8. Such differences may be due to the longer treatment at 42°C (3 h versus 30 min), but a proper control using the *TaPIF4D* protein would here be helpful to corroborate this regulation.

Response 13. According to the reviewer's suggestions, we conducted additional experiments to address the concern regarding the stability of *TaPIF4* protein. Specifically, we extended the duration of the in vitro protein degradation assay to 60

minutes at 42°C. The results demonstrated that the TaPIF4 protein exhibited significant degradation in the absence of TaSG-D1^{E286K} protein, with only approximately 7% of the protein remaining. In contrast, when TaSG-D1 or TaSG-D1^{E286K} was present, the percentage of retained TaPIF4 protein increased to 25% and 52%, respectively (new **Supplementary Figure 10d**). Furthermore, we included TaPIF4D (the phosphorylation-mimic form) as a control in vivo and vitro degradation assay, as suggested by the reviewer (new **Supplementary Figure 10d and Supplementary Figure 11b**), and the results showed consistency with the observations in the presence of TaSG-D1^{E286K}.

These additional experiments have reinforced our findings and provide further evidence for the role of TaSG-D1^{E286K} in enhancing the stability of TaPIF4 protein. We have incorporated these results into the revised manuscript to strengthen our conclusions in **line 182-183 and line 191-192, page 6**.

Reviewer #3 (Remarks to the Author):

The authors have addressed all my concerns.

We sincerely appreciate the reviewer's valuable assistance in improving the manuscript

Reviewer #4 (Remarks to the Author):

The authors have done nice job to address my concerns. However, one more question and one suggestion for the authors:

Comment 1: The author provided the information of the QTL interval, in which 292 genes are identified. What about the internal of the NILs (heterozygous region)? This is more promising.

Response: The genetic interval of the NILs is ~11.3 Mb in length, and contains 84 high-confidence genes, among which, five genes process variations in coding sequence including the *TaSG-D1* (**Supplementary Table 1**). We added the information in **page 3 line 68-71** in the revised manuscript.

Comment 2: Please involve the information from the rebuttal (e.g.the semi-dwarf phenotype of *pif4* and *TasgD1*) in the revised version.

Response: According to the reviewer's suggestion, we have added corresponding discussion in **line 358-367, page 11-12** into the revised manuscript as follows “The discrepancy suggests that other TaSG-D1-dependent regulators are involved in the regulation of plant height in wheat. Consistent with our hypothesis, a recent study demonstrated that TaSG-D1^{E286K} phosphorylates and stabilizes DELLA protein Rht-B1b, the Green Revolution gene in wheat, and the Rht-B1b protein then interacts with TaPIF4 and inhibits its transcriptional activity during plant development, and finally reduces plant height in wheat⁵. Moreover, it has been reported that DELLA proteins interact with BZR1 and suppress its transcriptional activity, leading to growth inhibition in *Arabidopsis*¹³. Therefore, even though the stability of TaPIF4 protein is enhanced in TaSG-D1^{E286K} lines, its transcriptional activity is inhibited, resulting in decreased plant height.”

References in response

1. Kim, T.-W. et al. Mapping the signaling network of BIN2 kinase using TurboID-mediated biotin labeling and phosphoproteomics. *The Plant Cell* **35**, 975–993 (2023).
2. Chen, X. et al. ERF49 mediates brassinosteroid regulation of heat stress tolerance in *Arabidopsis thaliana*. *BMC Biol* **20**, 254 (2022).
3. Li, J. et al. The GSK3-like Kinase BIN2 Is a Molecular Switch between the Salt Stress Response and Growth Recovery in *Arabidopsis thaliana*. *Developmental Cell* **55**, 367–380.e6 (2020).
4. Ye, K. et al. BRASSINOSTEROID-INSENSITIVE2 Negatively Regulates the Stability of Transcription Factor ICE1 in Response to Cold Stress in *Arabidopsis*. *The Plant Cell* **31**, 2682–2696 (2019).
5. Dong, H. et al. GSK3 phosphorylates and regulates the Green Revolution protein Rht-B1b to reduce plant height in wheat. *The Plant Cell* **35**, 1970–1983 (2023)
6. Ibañez, C. et al. Brassinosteroids Dominate Hormonal Regulation of Plant Thermomorphogenesis via BZR1. *Current Biology* **28**, 303–310.e3 (2018).
7. Kim, S. et al. The epidermis coordinates thermoresponsive growth through the phyB-PIF4-auxin pathway. *Nat Commun* **11**, 1053 (2020).
8. De Lucas, M. et al. A molecular framework for light and gibberellin control of cell elongation. *Nature* **451**, 480–484 (2008).
9. Koini, M. A. et al. High Temperature-Mediated Adaptations in Plant Architecture Require the bHLH Transcription Factor PIF4. *Current Biology* **19**, 408–413 (2009).
10. Sun, J., Qi, L., Li, Y., Chu, J. & Li, C. PIF4-Mediated Activation of YUCCA8 Expression Integrates Temperature into the Auxin Pathway in Regulating *Arabidopsis* Hypocotyl Growth. *PLoS Genet* **8**, e1002594 (2012).
11. Quint, M. et al. Molecular and genetic control of plant thermomorphogenesis. *Nature Plants* **2**, 15190 (2016).
12. Bernardo-García, S. et al. BR-dependent phosphorylation modulates PIF4 transcriptional activity and shapes diurnal hypocotyl growth. *Genes Dev.* **28**, 1681–1694 (2014).
13. Li, Q.-F. et al. An interaction between BZR1 and DELLAs mediates direct signaling crosstalk between brassinosteroids and gibberellins in *Arabidopsis*. *Sci Signal* **5**, ra72 (2012).

Reviewers' Comments:

Reviewer #1:

Remarks to the Author:

The authors have addressed all my concerns

Reviewer #2:

Remarks to the Author:

Authors addressed in this revised MS all points issued in my previous review and discussed the possible mechanisms for the unexpected observations. Overall, the MS discovers a novel role of TaPIF4 in heat-stress tolerance which I am convinced will be conserved in other plant species.

Reviewer #4:

Remarks to the Author:

The authors have successfully addressed my concerns.